



# JRAero: the Japanese Reanalysis for Aerosol v1.0

Keiya Yumimoto[1,*], Taichu Y. Tanaka[1], Naga Oshima[1], Takashi Maki[1]

[1]Meteorological Research Institute, Japan Meteorological Agency, Tsukuba, Ibaraki 305-0052, Japan
* Now at Research Institute for Applied Mechanics, Kyushu University, Kasuga-city, Fukuoka 816-8580, Japan

*Correspondence to*: Keiya Yumimoto (yumimoto@riam.kyushu-u.ac.jp)

**Abstract.** A global aerosol reanalysis product named the Japanese Reanalysis for Aerosol (JRAero) was constructed by the Meteorological Research Institute (MRI) of the Japan Meteorological Agency. The reanalysis employs a global aerosol transport model developed by MRI and a 2-dimensional variational data assimilation method. It assimilates maps of aerosol optical depth (AOD) from MODIS onboard Terra and Aqua satellites every 6 hours and has a TL159 horizontal resolution

(approximately $1.1° \times 1.1°$). This paper describes the aerosol transport model, the data assimilation system, the observation data, and the set-up of the reanalysis and examines its quality.

Comparisons with MODIS AODs showed that the reanalysis showed much better agreement than the free run (without assimilation) of the aerosol model and improved under- and overestimation in the free run, thus confirming the accuracy of the data assimilation system. The reanalysis had a root mean square error (RMSE) = 0.05, a correlation coefficient ($R$) = 0.96,

a mean fractional error (MFE) = 23.7%, a mean fractional bias (MFB) = 2.8%, and an index of agreement (IOA) = 0.98. The better agreement of the first guess, compared with the free run, indicates that aerosol fields obtained by the reanalysis can improve short-term forecasts.

AOD fields from the reanalysis also agreed well with monthly averaged global AODs obtained by the Aerosol Robotic Network (AERONET) (RMSE = 0.08, $R$ = 0.90, MFE = 28.1%, MFB = 0.6%, and IOA = 0.93). Site-by-site comparison

showed that the reanalysis was considerably better than the free run; RMSE was <0.10 at 86.4% of the 181 AERONET sites, $R$ was >0.90 at 40.7% of the sites, and IOA was >0.90 at 43.4% of the sites. However, the reanalysis tended to have a negative bias at urban sites (in particular, megacities in industrializing countries) and a positive bias at mountain sites, possibly because of insufficient anthropogenic emissions data, the coarse model resolution, and the difference in representativeness between satellite and ground-based observations.





## 1 Introduction

Airborne aerosols are tiny particles that are globally distributed in Earth's atmosphere from the troposphere to the stratosphere (including over urban areas, oceans, deserts, and forests), and that affect various aspects of human society and the Earth system. In the short-term and most obviously, aerosol particles can degrade visibility and damage aviation and transport (Wilkinson et al., 2012; Prata and Tupper, 2009). The World Health Organization (WHO), which has reported country estimates of air pollution exposure and its health impact (WHO, 2016), has suggested that 6.5 million deaths (11.6% of all global deaths) may be associated with indoor or outdoor air pollution, and 92% of the world's population lives in places where air quality levels do not meet the WHO Ambient Air quality guideline of an annual mean $PM_{2.5}$ (particulate matter with a diameter of less than 2.5 μm) concentration of less than 10 μg/m$^3$. Impacts on the Earth system include effects on ocean biogeochemistry and climate. For instance, 500 Mt of dust particles settles to the oceans and supplies iron, a vital element for ocean productivity (Shao et al., 2011). Aerosols also modify the radiation balance and influence the albedo and properties of clouds through both direct and indirect effects (IPCC, 2013).

In recent decades, our understanding of the aerosol life cycle and aerosol impacts has been advanced by innovation and progress in both observational and numerical modeling techniques. Among various observation techniques, remote sensing using passive and active sensors has revealed both the spatial distribution and temporal evolution of aerosols at regional and global scales. The Moderate Resolution Imaging Spectroradiometer (MODIS) onboard the Terra and Aqua satellites is an example of a passive sensor. MODIS has provided aerosol optical properties (AOPs), including aerosol optical depth (AOD) and the Ångström exponent, globally since 2000 (Remer et al., 2005; Levy et al., 2007; Zhang and Reid, 2006). More recently, Himawari-8, a geostationary meteorological satellite launched on 7 October 2014, has been providing full-disk images of AOPs every 10 minutes (Bessho et al., 2016; Kikuchi et al., 2016; Yumimoto et al., 2016). The Aerosol Robotic Network (AERONET; http://aeronet.gsfc.nasa.gov/) is a global ground-based network of passive instruments, including sun photometers, that has provided AOPs at several wavelengths for more than 20 years (Holben et al., 1998, 2001; Li et al., 2014). Backscatter lidar is an active-type aerosol observation sensor. The Cloud-Aerosol Lidar with Orthogonal Polarization (CALIPSO), the first polarization lidar in orbit, has provided continuous global measurements of vertical aerosol distributions since 2006 (Winker et al., 2010; Liu et al., 2008). The European Aerosol Research Lidar Network (EARLINET; https://www.earlinet.org; Matthais et al., 2004) and the Asian Dust Network (AD-Net; http://www-lidar.nies.go.jp/AD-Net; Sugimoto et al., 2008) are regional ground-based lidar networks, and the Micro-Pulse Lidar Network (MPLNET; http://mplnet.gsfc.nasa.gov) is a global ground-based lidar network. Although satellites and ground-based networks have remarkably increased the amount of observed data that are available, observational coverage is still limited and spatially and temporally uneven. Passive satellite-borne sensors cover wide swaths across large regions, but their retrievals are usually vertically integrated AOPs that lack information on vertical distributions, and they can be obtained only under daytime and clear-sky conditions. Active sensors have the capability of measuring vertical aerosol profiles both under clouds (by ground-



based lidar) and above clouds (by space-based lidar). However, their field of view is quite narrow compared with that of passive sensors so it is difficult to use lidar to cover large regions.

Numerical modeling of the aerosol life cycle has advanced considerably in the last decade. Various global aerosol transport models have been developed, and multimodel intercomparison projects have been carried out (e.g., AeroCom, Kinne et al., 2006; and International Cooperative for Aerosol Prediction [ICAP], Sessions et al., 2015). Global aerosol transport models have been developed by numerous weather prediction centers, research institutes, and universities (Kinne et al., 2006; Sessions et al. 2015 and references therein). An aerosol transport model can provide three-dimensional, gridded aerosol distribution data at regular time intervals. However, multimodel intercomparison projects have shown that large differences and uncertainties due to insufficient emissions data, poor parameterization of aerosol processes (e.g., transport, chemistry, settling, and deposition), and errors in meteorological fields remain in the models.

Data assimilation, which is the integration of observation data into a numerical model, is one way of overcoming these shortcomings. At an early stage in the development of data assimilation methods, Collins et al. (2001) and Wang et al. (2006) assimilated satellite measurements by using an optimal interpolation method and a Newtonian nudging scheme. Hakami et al. (2005) and Yumimoto et al. (2007, 2008) attempted to apply a four-dimensional variational method (a so-called advanced data assimilation method) to inverse modeling of black carbon (BC) and dust aerosols with ground-based observations and regional models. To date, measurements obtained by various observation platforms, including MODIS (Dai et al., 2014; Huneeus et al., 2012; Wang et al., 2012; Zhang et al., 2008), CALIPSO (Sekiyama et al., 2010; Zhang et al., 2011, 2014), Himawari-8 (Yumimoto et al., 2016), AERONET (Schutgens et al., 2010a), and surface $PM_{10}$ (particulate matter with diameters less than 10 μm) monitoring systems (Tombette et al., 2009; Lee et al., 2013; Jiang et al., 2013), have been used in assimilation studies adopting both variational (Benedetti et al., 2009; Dubovik et al., 2008; Hakami et al., 2007; Henze et al., 2007; Yumimoto and Takemura, 2013) and ensemble-based (Rubin et al., 2016; Schutgens et al., 2010b; Di Tomaso et al., 2017; Yumimoto and Takemura, 2011) methods.

One of most important outcomes of data assimilation is the generation of uniform, continuous, and systematic best-estimated data products (i.e., reanalysis products) for use by the research community. Several weather prediction centers, including the Japan Meteorological Agency (JMA), the European Centre for Medium-Range Weather Forecasts (ECMWF), the U.S. National Center for Atmospheric Research/National Centers for Environmental Prediction (NCAR/NCEP), and the Global Modeling and Assimilation Office (GMAO) of NASA have developed meteorological reanalysis products that are widely utilized by the research community (Kobayashi et al. 2015; Harada et al., 2016; Uppala et al., 2005; Dee et al., 2011; Kalnay et al., 1996; Rienecker et al., 2011). In addition to these meteorological reanalysis products, aerosol reanalysis products are under development. ECMWF has generated a global reanalysis data set of atmospheric composition including Carbon Monoxide, ozone and aerosols for the period 2003–2015 known as the Copernicus Atmosphere Monitoring Service (CAMS) interim reanalysis (CAMSiRA) (Inness et al., 2013; Flemming et al., 2017). GMAO NASA has provided an aerosol reanalysis product called the Modern-Era Retrospective analysis for Research and Applications, Version 2 (MERRA-2; Buchard et al., 2015). The U.S. Naval Research Laboratory (NRL) has developed an 11-year global gridded aerosol



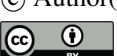

reanalysis product based on the NAVDAS-AOT assimilation system with the NRL Aerosol Analysis and Prediction System (NAAPS; Lynch et al., 2016). Aerosol reanalysis products have various applications, including determination of the initial and boundary conditions of not only aerosol transport models but also numerical weather forecasting models, climatological analyses of aerosols and their climate effect, satellite measurement retrievals (e.g., Yokota et al., 2009), use as truth data for

observation system simulation experiments (e.g., Yumimoto, 2013; Zoogman et al., 2014), and use as input data for epidemiologic studies of $PM_{2.5}$ (e.g., Atkinson et al., 2014; Kloog et al., 2011). However, at this time, aerosol reanalysis is still at an early stage of development.

    In this study, we conducted a global aerosol reanalysis named the Japanese Reanalysis for Aerosol (JRAero) for the period 2011–2015. The valued-added MODIS AOD observations provided by the NRL and the University of North Dakota (NRL-

UND MODIS AODs) were assimilated into the MASINGAR mk-2 (Model of Aerosol Species IN the Global AtmospheRe), a global aerosol transport model developed at JMA's Meteorological Research Institute (MRI), by using a two-dimensional variational data assimilation system. This paper constitutes a comprehensive report on JRAero. The aerosol model, the data assimilation method, and the observation data used in this study are outlined in Sects. 2.1–2.3, and the set-up of the reanalysis is presented in Sect. 2.4. Section 3 focuses on the evaluation of the reanalysis product with independent

observation data. We present our conclusions in Sect. 4, and we offer remarks on future directions in the development of this reanalysis in Sect. 5.

## 2 Description of the data assimilation system

The data assimilation system consists of an aerosol model, a data assimilation module, and observation data. We describe each component in the following subsections. The setup of the reanalysis and the observation dataset used for an

independent validation are also presented.

### 2.1 Overview of MASINGAR mk-2

The global aerosol transport model MASINGAR mk-2 (Yukimoto et al., 2012) is the first major update to the original MASINGAR (Tanaka et al., 2003), which was developed by MRI and JMA. MASINGAR mk-2 is coupled online with an atmospheric general circulation model (AGCM; MRI-AGCM3; Yukimoto et al., 2012) through a general-purpose coupler

(Scup: Yoshimura and Yukimoto, 2008), and it is capable of treating the major tropospheric aerosol components, BC, organic carbon (OC), mineral dust, sea-salt, and sulfate aerosols, and their precursors. JMA started to use the original MASINGAR for operational sand and dust storm forecasting in January 2004, and it changed over to MASINGAR mk-2 in November 2014. MASINGAR mk-2 serves as a member of the ICAP multi-model ensemble (MME) (Sessions et al., 2015). The main features of the update were (1) replacement of the coupled AGCM, (2) improvement in the treatment of

stratospheric sulfate aerosols, (3) renovation of emission schemes for dust and sea-salt aerosols, and (4) redesign of calculation of AOPs. Here, we provide an overview of MASINGAR mk-2 in which we focus on the updated features.





### 2.1.1 Physical processes

MASINGAR mk-2 includes advection, convective, diffusive transport, emissions, chemical reaction, and removal processes. In the model, the continuity equation of the volume mixing ratio of the $i$-th aerosol component at the time step $n+1$ ($x_i(t_{n+1})$) is solved by successively applying operators, as follows:

$$x_i(t_{n+1}) = A \cdot D \cdot C \cdot T \cdot x_i(t_n) \,, \tag{1}$$

where $A$, $D$, $C$, and $T$ denote operators associated with advection, eddy diffusion, convective transport, and transformations, respectively, due to emissions, depositions, and chemical reactions.

The model employs a semi-Lagrangian advection scheme (Staniforth and Côtê, 1991) for advection ($A$). The scheme allows the use of a much longer time step compared with an Eulerian advection scheme while conserving numerical stability and accuracy. In the scheme, the upstream point is first searched with horizontal and vertical wind fields and then the mixing ratio is obtained by a three-dimensional interpolation. The interpolation includes a correction for overshooting or undershooting the advection fluxes to ensure mass conservation and a non-negative value.

Eddy diffusion ($D$) is defined as

$$\left(\frac{\partial x_i}{\partial t}\right)_{\text{diffusion}} = \frac{\partial}{\partial z}\left(K_z \frac{\partial x_i}{\partial z}\right) \tag{2}$$

where $K_z$ is the vertical eddy diffusion coefficient. The coefficient for aerosols is assumed to be the same as that for water vapor and is provided by MRI-AGCM3 through the coupler. MRI-AGCM3 calculates the diffusion coefficient by the turbulence model of Nakanishi and Niino (2006, 2009), which is an improved version of the MY scheme (Mellor and Yamada, 1974, 1982) called the MYNN scheme. Convective transport ($C$) is estimated by using updraft or downdraft mass fluxes calculated by the mass-flux-type cumulus parameterization scheme developed by Yoshimura et al. (2015) and embedded in MRI-AGCM3. Further details of convective transport are given by Yukimoto et al. (2012).

The transformation process ($T$) includes gravitational settling, dry and wet depositions, emissions, and chemical reactions. Gravitational settling is estimated from the terminal velocity ($V_s$), which is estimated on the basis of Stokes' law as follows:

$$V_s = \frac{2C_c(\rho_p - \rho_a)gr_p^2}{9\mu} \,, \tag{3}$$

where $Cc$ is the Cunningham slip-flow correction, $\rho_a$ is air density, $\rho_p$ and $r_p$ are the density and radius of the aerosol particles, $g$ is gravitational acceleration, and $\mu$ is the viscosity of air. The model assumes that the particles are spherical. Ginoux (2003) investigated the effect of nonsphericity on gravitational settling and suggested that the introduction of nonsphericity to the modeling of gravitational settling does not significantly improve dust modeling. Dry deposition at the ground surface removes aerosol particles from the atmosphere. To calculate the dry deposition velocity ($V_d$), the model employs the resistance analog model (Seinfeld and Pandis, 2016), in which the dry deposition velocity is expressed as the inverse of the sum of resistances,





$$V_d = \frac{1}{r_a + r_b + r_a r_b V_s},$$ (4)

where $r_a$ and $r_b$ are the aerodynamic and quasi-laminar resistance, respectively. Tanaka et al. (2003) have described derivation of the resistances in detail.

MASINGAR mk-2 considers two wet deposition processes, namely, in-cloud and below-cloud scavenging. In-cloud
scavenging is parameterized following Giorgi and Chameides (1986), and the loss rate $\Lambda_{ic}$ in $s^{-1}$ is estimated as

$$\Lambda_{ic} = \frac{F_c(1 - \exp(\beta T_c))}{\Delta t},$$ (5)

where $F_c$, $T_c$, $\beta$, and $\Delta t$ are the fraction of the cloud with precipitation, the duration of the precipitation, the frequency of conversion of cloud water to rainwater, and the model time step, respectively. These parameters are derived from the rainwater formation rate and the cloud amount provided by the AGCM. The loss rate due to below-cloud scavenging $\Lambda_{bc}$ is
evaluated as

$$\Lambda_{bc} = \frac{\lambda_{bc} P}{\Delta t},$$ (6)

where $\lambda_{bc}$ and $P$ are the below-cloud scavenging coefficient and precipitation intensity, respectively. To obtain $\lambda_{bc}$, the collision efficiency between aerosol particles and raindrops is calculated by considering Brownian diffusion, interception, and inertial impaction (Slinn, 1984; Seinfeld and Pandis, 2016). Tanaka and Chiba (2005) describe the below-cloud
scavenging procedure in detail. Gravitational settling and dry and wet deposition of mineral dust and sea-salt aerosols are particle size-dependent. The model also includes a process for the reemission of particles due to evaporation of raindrops. The fraction of reemitted particles is assumed to be proportional to the amount of evaporated rainwater.

### 2.1.2 Aerosol components

The standard version of MASINGAR mk-2 represents aerosols with 10 externally mixed size distributions (i.e., mineral dust,
sea salt, and five carbonaceous and three sulfate categories). The model represents mineral dust and sea-salt aerosols by discrete size bins and assumes lognormal size distributions for the other aerosol components.

For mineral dust particles, the model uses a size bin method that logarithmically divides the particle size range from 0.2 to 20 µm into 10 size classes. The volumetric mean radii of dry particles in each size bin are 0.136, 0.215, 0.340, 0.540, 0.855, 1.355, 2.148, 3.405, 5.396, and 8.553 µm. The dust particle density is assumed to be 2.65 g cm$^{-3}$ following Tegen and Fung
(1994). The dust emission scheme was completely replaced in the update. Now, the dust emission flux is estimated by using the wind erosion model developed by Shao et al. (1996) instead of by using the empirical approach of Gillette (1978). The dust emission flux from the $i$-th dust size bin $F_i$ is estimated as

$$F_i = C_d A F_{0i},$$ (7)



where $C_d$, $A$, and $F_{0i}$ are a global tuning parameter, the erodible area fraction, and the dust emission flux, respectively, estimated by the wind erosion model. $C_d$ is set to 0.001 based on the study by Tanaka and Chiba (2005). The erodible area fraction depends on ground-surface conditions and is represented by

$$A = (1 - A_v)(1 - A_s)(1 - A_w)A_l A_t \, , \qquad (8)$$

where $A_v$, $A_s$, $A_w$, $A_l$, and $A_t$ denote the land-cover factors vegetation, snow, and water, land use, and soil type, respectively. Equation (8) means that larger cover with vegetation, snow, or water suppresses dust emissions. Vegetation cover $A_v$, is a function of the leaf area index (LAI) and is set to 1 when LAI is larger than a threshold value (1.2), following Lunt and Valdes (2002). The area fraction of snow cover is used for $A_s$. Thus, when the whole area of a model grid is covered by snow, the model estimates zero dust emissions from that grid. The value of LAI and the area fraction of snow cover are provided

by Hydrology, Atmosphere and Land (HAL), the land surface model embedded in MRI-AGCM3. The area fraction of inland water (i.e., oceans, rivers, and lakes) is used for water cover $A_w$, and dust emission from grids covered by inland waters is suppressed. The land-use type is used to identify potential erodible surfaces. For instance, a grid covered by "evergreen broadleaf forests" is not a potential erodible surface, so its land-use factor $A_l = 0$, whereas grids covered by "sand desert" have $A_l = 1$. The Land Class Type (LCT) database provided by the U.S. Geological Survey (USGS, http://www.usgs.gov) is

used to obtain $A_w$ and $A_l$. Surfaces covered by Lithosol are excluded as a possible dust source (i.e., $A_t = 0$), and soil texture data from the Food and Agriculture Organization (FAO) are used to calculate $A_t$.

The wind erosion model estimates the friction velocity, threshold friction velocity, and saltation flux and then calculates the dust emission flux $F_{0i}$. The saltation flux of a saltating particle of diameter $D_s$ is calculated as follows:

$$Q(D_s) = \begin{cases} \frac{C_s(D_s)\rho_a u_{*s}^3}{g}\left(1 - \frac{u_{*t}(D_s)^2}{u_{*s}^2}\right), \ u_{*s} > u_{*t} \\ 0, \ u_{*s} \le u_{*t} \end{cases} . \qquad (9)$$

Here, $u_{*t}(D_s)$ is the threshold friction velocity of a saltating particle of diameter $D_s$, and $u_{*s}$ is the friction velocity. $Cs$ is a coefficient that depends on the saltating particle of diameter $D_s$. The threshold friction velocity is calculated using the empirical formula of Shao and Lu (2000), and the effects of soil water are calculated in accord with Fécan et al. (1999). To consider the frozen soil effect (Kang et al., 2013), the soil water factor of Fécan et al. (1999) is enhanced by 50% when the soil temperature drops below 0°C. The roughness length for momentum transfer and soil water content provided by HAL are

used for the estimation of $u_{*t}(D_s)$ and $u_{*s}$. Finally, the wind erosion model calculates the dust emission flux $F_{0i}$ from the saltation flux using the energy-based dust emission scheme proposed by Shao et al. (1996):

$$F_{0i} = \beta \, Q \, u_{*t,d}^{-2} \qquad (10)$$

Here, $u_{*t, d}$ is the threshold friction velocity of a dust particle, and $\beta$ is an empirical function of the particle diameter of a saltating particle ($d_s$) and a dust particle ($d_d$):

$$\beta = 10^{-5}[1.25 \ln(d_s) + 3.28]\exp(-140.7d_d + 0.37). \qquad (11)$$

Tanaka and Chiba (2005) give further details of the wind erosion model.



MASINGAR mk-2 has 10 size bins for sea-salt aerosols. The volumetric mean radii of the size bins are the same as those for mineral dust aerosols. The particle density is set to 2.25 g cm⁻³ (Hänel, 1976). The original MASINGAR employed the following empirical formula developed by Monahan et al. (1986) for the density function (particles m⁻² s⁻¹ µm⁻¹):

$$\frac{dF}{dr} = 1.373u_{10}^{3.41}r^{-A}(1 + 0.057r^C)\times10^{De^{-B^2}} ,$$ (12)

where $u_{10}$ is the wind speed at 10 m above ground level, $r$ is the particle radius at relative humidity (RH) = 80%, $A = -3$, $B = (0.380 - \log(r))/0.650$, $C = 1.05$, and $D = 1.19$. In MASINGAR mk-2, the settings $A = 4.7(1 + \Theta r)^{-0.017r^{-1.44}}$(where $\Theta$ is an adjustable parameter describing the shape of the sub-micron particle size distribution; here $\Theta = 30$ is used), $B = (0.433 - \log(r))/0.433$, $C = 3.45$, and $D = 1.607$ are used in Eq. (12) following suggestions by Gong (2003).

Carbonaceous aerosols are classified as BC or OC, and BC and OC are further divided into hydrophobic or hydrophilic
states. It is assumed that 80% of BC and 50% of OC are emitted from both anthropogenic sources and biomass burning in a hydrophobic state, and the rest in a hydrophilic state. Hydrophobic BC and hydrophobic OC both become hydrophilic through aging processes, and the conversion rate is based on an e-folding time of 1.2 days (Chin et al., 2002). MASINGAR mk-2 includes OC production from terpene. OC particles produced from terpene are treated as hydrophilic and terpene emissions data are used to obtain the amount of OC from terpene. It is assumed that only hydrophilic species are removed by
wet deposition processes. The size distributions of BC and OC are represented by lognormal distributions with a number-equivalent geometric mean radius of 0.0118 and 0.0212 µm, respectively, and standard deviation of 2.0 and 2.2, respectively, under dry conditions (Hess et al., 1998; Chin et al., 2002). The particle density of BC is assumed to be 1.25 g cm⁻³, and that of OC is assumed to be 1.8 g cm⁻³.

MASINGAR mk-2 has a sulfur chemistry model that treats eight major sulfur compounds, including sulfur dioxide (SO₂),
sulfate ($SO_4^{2-}$), dimethyl sulfide (DMS), hydrogen sulfide (H₂S), carbonyl sulfide (OCS), and methane sulfonic acid (MSA), and it includes seven gas-phase reactions and two aqueous-phase reactions. Tanaka et al. (2003) present details of the sulfur chemistry model. It is well known that OCS contributes to the formation of stratospheric sulfate aerosols (Turco et al., 1980). The chemical process including OCS was newly added to MASINGAR mk-2 to improve simulation of stratospheric sulfate aerosols. MASINGAR mk-2 can separately treat sulfate aerosols originating from anthropogenic, biogenic, and volcanic
sources. The size distribution of the sulfate aerosols is also represented by a lognormal size distribution with a mean radius of 0.07 µm and a standard deviation of 2.03 under dry conditions (Hess et al., 1998; Chin et al., 2002), although different size distributions can be given for each sulfate component. The particle density of sulfate is assumed to be 1.7 g cm⁻³.

MASINGAR mk-2 receives meteorological fields (e.g., wind fields, air temperature, surface pressure, clouds, precipitation, and the eddy diffusion coefficient) and surface conditions (e.g, soil water content and LAI) from the AGCM for the
calculation of the advection, diffusion, deposition, and emission processes. In turn, the AGCM receives mass concentrations and deposition fluxes of the aerosol components from MASINGAR mk-2 via the coupler for the simultaneous calculation of





the direct and indirect radiative effects of aerosols in its simulation. Yukimoto et al. (2012) provide a detailed description of the calculation of radiative effects.

### 2.1.3 Calculation of aerosol optical depth

The extinction coefficient at a given wavelength of aerosol component $l$ in the $k$-th vertical layer is calculated as follows (e.g., Tegen and Lacis, 1996; Chin et al., 2002):

$$\alpha_{l,k} = \frac{3}{4} \frac{Q_l x_{l,k}}{\rho_{a\,l} r_{\mathrm{eff}\,l}}. \tag{13}$$

Here, $Q_l$ is the extinction efficiency factor, which is the cross section-weighted mean extinction efficiency over a given particle size distribution of the $l$-th aerosol component, $x_{l,k}$ denotes the mass concentration of the $l$-th aerosol component in the $k$-th vertical layer, $\rho_{a\,l}$ is the particle mass density of the $l$-th aerosol component, and $r_{\mathrm{eff}\,l}$ is the effective radius (cross section-weighted mean radius over a given particle size distribution) of the $l$-th aerosol component. $Q_l$ is calculated on the basis of Mie scattering theory for a homogeneous sphere at a given wavelength using the modeled aerosol size distributions (see Sect. 2.1.2) and the complex refractive index, obtained from the software package Optical Properties of Aerosols and Clouds (Hess et al., 1998). Under humid conditions, all quantities of hygroscopic components in Eq. (13) change with RH because they take up water. The factors for hygroscopic growth with RH of the hygroscopic components (i.e., sulfate, hydrophilic OC and BC, and sea-salt aerosols) are taken from Chin et al. (2002). The complex refractive indices of the hygroscopic components are determined by volume-weighted averaging of RH and the complex refractive index of each dry component. In the calculation of the extinction coefficient, the sulfate component is assumed to be ammonium sulfate and the mass concentration of the sulfate component is increased by the ammonium sulfate-to-sulfate molecular ratio to compensate for the absence of ammonium in the model. The extinction coefficient of organic aerosols is estimated by replacing the OC mass concentration with the organic matter (OM) mass concentration using an OM-to-OC factor of 1.4 (White and Roberts, 1977; Japar et al., 1984; Russell, 2003). MASINGAR mk-2 calculates the aerosol extinction coefficients at wavelengths of 550 and 870 nm. The total aerosol optical depth (AOD) is derived by integration of $\alpha_{l,k}$ in all aerosol components and all model vertical layers as follows:

$$\tau = \sum_{l=1}^{L} \sum_{k=1}^{K} \alpha_{l,k} \Delta z_k. \tag{14}$$

Here, $L$, $K$, and $\Delta z_k$ are the number of aerosol components, the number of model vertical layers, and the box height between the upper and lower boundaries of the $k$-th vertical layer, respectively. We used the modeled AOD value at the typical visible wavelength of 550 nm in this study.

### 2.2 Assimilation method

We previously developed an aerosol data assimilation system (MASINGAR/LETKF) based on MASINGAR mk-2 and an ensemble-based assimilation technique called the Local Ensemble Transform Kalman Filter (LETKF; Hunt et al., 2007;



Miyoshi and Yamane, 2007). In fact, Yumimoto et al. (2016) have reported successful results with MASINGAR/LETKF in the assimilation of products from MODIS and Himawari-8. However, the computational cost of MASINGAR/LETKF is quite high owing to the necessity of ensemble simulation, and it is still unrealistic for development of the long-term reanalysis product. Therefore, in this study, for the initial development of a reanalysis product we developed an aerosol data

assimilation system based on a sequential variational method. The assimilation system (MASINGAR/2D-Var) uses a 2-dimensional variational method (2D-Var) for the core of the assimilation system. NAVDAS-AOT (Zhang et al., 2008, 2014) and the NAAPS reanalysis also employ 2D-Var (Lynch et al., 2016). Experience and knowledge obtained during the development of MASINGAR/LETKF, and by conducting experiments with that system, were utilized in the development of MASINGAR/2D-Var.

The cost function in a 3-dimensional variational method (3D-Var) is generally defined as

$$J_x(x) = \frac{1}{2}(x - x^f)^T \mathbf{P}^{-1}(x - x^f) + \frac{1}{2}(y^o - H(x))^T \mathbf{R}^{-1}(y^o - H(x)),$$ (15)

where $x$ denotes the vector of modeled aerosol mass mixing ratios. The suffix $f$ represents the forecast (*a priori*), and $y^o$ denotes a vector that contains observations used for the assimilation. $H$ is an observation operator that transforms model variables to observation space. In this study, we used the NRL-UND MODIS AODs ($\tau^o$) as the observational constraint ($y^o$).

Therefore, the observation operator includes the conversion of the aerosol mass mixing ratio into AODs (described in Sect. 2.1.3) and the interpolation of model space into observation space. The first term on the right-hand side of Eq. (15) represents the departure of the analysis (*a posteriori*) aerosol mass mixing ratio from the forecast ratio weighted by the background error covariance matrix for the aerosol mass mixing ratio (**P**). The second term on the right-hand side represents the difference between the modeled and observed AODs weighted by the observation error covariance matrix (**R**). To search

for the optimal solution, which minimizes the cost function, the gradient vector of Eq. (15) is calculated as follows:

$$\nabla J_x = \mathbf{P}^{-1}(x - x^f) - \mathbf{H}^T \mathbf{R}^{-1}(y^o - H(x)).$$ (16)

Here, $\mathbf{H}^T$ is an adjoint of $H$.

For the assimilation of 2-dimensional observations such as the NRL-UND MODIS AODs, we reduced the computational cost by degrading the assimilation system from 3-dimensional to 2-dimensional and analyzed the modeled AOD (2-

25 dimensional variable) instead of the modeled aerosol mixing ratio (3-dimensional variable). The cost function (Eq. (15)) is redefined as

$$J_\tau(\tau) = \frac{1}{2}(\tau - \tau^f)^T \mathbf{P}_\tau^{-1}(\tau - \tau^f) + \frac{1}{2}(\tau^o - \mathbf{H}_I \tau)^T \mathbf{R}^{-1}(\tau^o - \mathbf{H}_I \tau),$$ (17)

where $\tau$ denotes the modeled AOD vector, and $\mathbf{P}_\tau$ is the background error covariance matrix for AOD. $\mathbf{H}_I$ is the interpolation into observation space and a linear operator. The gradient vector of Eq. (16) is derived as follows:

$$\nabla J_\tau = \mathbf{P}_\tau^{-1}(\tau - \tau^f) - \mathbf{H}_I^T \mathbf{R}^{-1}(\tau^o - \mathbf{H}_I \tau).$$ (18)

Because the observation operator $\mathbf{H}_I$ is a linear operator, the cost function (Eq. (17)) becomes a quadratic scalar function. Therefore, the minimum of the cost function is obtained for $\tau = \tau^a$, which makes $\nabla J_\tau = 0$. The analysis increment of AOD is derived from Eq. (18) as





$$\delta\boldsymbol{\tau}^a = \boldsymbol{\tau}^a - \boldsymbol{\tau}^f = \mathbf{K}(\boldsymbol{\tau}^o - \mathbf{H}_I\boldsymbol{\tau}^f), \tag{19}$$

Suffix $a$ represents the analysis (*a posteriori*); $(\boldsymbol{\tau}^o - \mathbf{H}_I\boldsymbol{\tau}^f)$ is the innovation; and $\mathbf{K}$ is the Kalman gain, defined as follows:

$$\mathbf{KK} = \mathbf{P}_\tau \ \mathbf{H}_I{}^T\left(\mathbf{R} + \mathbf{HP}_\tau \ \mathbf{H}_I{}^T\right)^{-1}. \tag{20}$$

Because the MODIS AOD is the product of a column-integrated aerosol optical property and does not provide any
information about the vertical profile or the aerosol components, we allocate the analysis increment of AOD to the aerosol
mass mixing ratio while keeping the shape of the vertical profile and the rate of each aerosol component in the forecast (*a priori*) aerosol fields. First, the analysis increment of AOD is derived for each aerosol component. The analysis increment of
the AOD for the *l*-th aerosol component is calculated as

$$\delta\boldsymbol{\tau}_l^a = \delta\boldsymbol{\tau}^a \frac{\tau_l^f}{\tau^f}, \tag{21}$$

where $\tau_l^f$ is the forecast AOD of the *l*-th aerosol component. Then we distribute the analysis increment of AOD of each
aerosol component to the aerosol mixing ratios in each vertical layer. The analysis increment of the mixing ratio of the *l*-th
aerosol component and the *k*-th vertical layer is derived as

$$\delta\boldsymbol{x}_{l,k}^a = \delta\boldsymbol{\tau}_l^a \frac{x_{l,k}^f}{\alpha_{l,k}^f} \frac{\alpha_{l,k}^f}{\tau^f}, \tag{22}$$

where $x_{l,k}^f$ and $\alpha_{l,k}^f$ represent the forecast mixing ratio and the extinction coefficient of the *l*-th aerosol component and the *k*-
th vertical layer, respectively. In this distribution, we assume that the hygroscopic growth rate is unchanged. Finally, we
obtain the analysis mixing ratio as follows:

$$\boldsymbol{x}_{l,k}^a = \boldsymbol{x}_{l,k}^f + \delta\boldsymbol{x}_{l,k}^a. \tag{23}$$

MASINGAR/2D-Var is developed with expansibility that allows the assimilation of 3-dimensional observations in the future
update.

We introduced a localization technique used in LETKF to the system that divides the model space into local regions using
a prescribed localization scale and performs the analysis calculation (Eq. (19)) independently. Use of this technique reduces
spurious error covariance with distance and enables parallel processing to be used to reduce computational cost. Zhang et al.
(2008) calculated the spatial correlation between satellite observations and model forecasts as a function of distance and
found that when the distance was more than 1000 km, the spatial correlation decreased to less than 0.05. On the basis of their
results, we set the localization scale to 1000 km.

We define the background error covariance between model grids $m$ and $n$ (i.e., the $(m, n)$ element of $\mathbf{P}\tau$) as

$$P_\tau^{m,n} = \sigma_\tau^m\left(C^{m,n}\right)^{1/2}\sigma_\tau^n, \tag{24}$$

where $\sigma_\tau^m$ and $\sigma_\tau^m$ are the background error standard deviations at model grids $m$ and $n$, respectively. $C^{m,n}$ is a smooth
weighting function whose value becomes smaller when the distance between model grids $m$ and $n$ becomes larger. The
function prevents a spurious large covariance between distant model grids. We use the second-order auto-regressive (SOAR)
approximation (Daley and Barker, 2001) following Zhang et al. (2008) to calculate value of $C^{m,n}$,



$$C^{m,n} = \left(1 + \frac{R^{m,n}}{L}\right) \exp\left(1 + \frac{R^{m,n}}{L}\right), \tag{25}$$

where $R^{m,n}$ denotes the distance (km) between model grids $m$ and $n$, and $L$ is the horizontal error correlation length, which we set to 200 km following Zhang et al. (2008).

The background error standard deviation was derived from the MASINGAR mk-2 simulation without data assimilation (free run). We collected AOD values within ±15 days of the targeted hour, and then calculated their mean value ($\bar{\tau}_{FR}$) and standard deviation ($\hat{\sigma}_{FR}$). For instance, for 12:00 UTC on 25 July 2014, AOD values at 12:00 UTC on 15 July, 16 July, 17 July, … 9 August were gathered (i.e., 31 AOD values) and used for the calculation of the mean and standard deviation. We calculated the background error standard deviation as follows:

$$\sigma_\tau = \frac{\hat{\sigma}_{FR}}{\bar{\tau}_{FR}} \tau^f. \tag{26}$$

Here, $\tau^f$ is the forecast AOD. The fraction on the right-hand side of Eq. (26) indicates that the standard deviation is normalized by the mean value. To introduce the flow-dependent structure of AOD into the background error covariance, the normalized standard deviation is multiplied by the forecast AOD.

### 2.3 Observation data

### 2.3.1 Assimilation dataset: the NRL-UND MODIS AODs

We employed the MODIS Level 3 AOD product provided by the NRL and UND (https://earthdata.nasa.gov/earth-observation-data/near-real-time/nrt-value-added-modis-aerosol-optical-depth-product-available) as the assimilation dataset. The NRL-UND MODIS AOD product was produced for the purpose of aerosol data assimilation and is based on the NASA operational MODIS Level 2 Collection 5 AOD dataset (Remer et al., 2005; Levy et al., 2007). These data (i.e., Dark Target AODs) have been subjected to extensive quality assurance (QA) and quality check (QC) procedures (Zhang and Reid 2006; Shi et al., 2011; Hyer et al., 2011). The QA and QC procedures included (1) a stringent filter to reduce outliers, eliminate cloud contamination, and exclude bad conditions, when aerosol detection is likely to be inaccurate, and (2) empirical corrections to reduce systematic biases over land and ocean. Then, the data are aggregated onto a 1° × 1° grid to reduce random spatial variation in AOD values with additional buddy checks. The product is provided every six hours (i.e., 0:00, 6:00, 12:00, 18:00 UTC). The QA and QC procedures and the validation of the processed data have been reported by Zhang and Reid (2006), Shi et al. (2011), and Hyer et al. (2011). Although the cutoff latitudes for the product are 40°S over water in the Southern Hemisphere and 80°N in the Northern Hemisphere, in this study, the AOD product observed from 60°S to 60°N was assimilated. We assumed that the observation error covariance matrix (**R**) was diagonal and assigned the error estimated by the QA and QC procedure to the observation error (i.e., the diagonal component of the observation error covariance matrix).

The horizontal distribution of the total number of NRL-UND MODIS AOD data assimilated in the reanalysis run (RA) of MASINGAR mk-2 during the entire reanalysis period and seasonally is shown in Fig. 1. Little or no observation data are

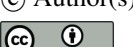



available from regions covered by bright deserts (e.g., Saharan Desert, Kalahari Desert of southern Africa, Arabian Peninsula, the Middle East, Central Asia, inland deserts of China, and the inland desert of Australia). Cloud coverage also affects the availability of observation data. For instance, the Intertropical Convergence Zone, which is frequently covered by clouds, has relatively fewer observation data. During winter in the Northern Hemisphere (i.e., December–February), little or

no observation data are available from higher latitudes because of snow or ice cover. There are no AOD observations over the ocean south of 40°S because of the cutoff used by the QA/QC procedure. This cutoff aims to filter out cloud contamination in the Southern Ocean (Toth et al., 2013). The time series of the number of the NRL-UND MODIS AOD data assimilated in RA shows that there are more observation data during the boreal summer than during the boreal winter (Fig. 2a). No observation data were available from 6:00 UTC on 1 October 2013 to 9:00 UTC on 9 October 2013. On average,

about 1400 observations were assimilated at each analysis time. We assimilated about 2,000,000–2,900,000 data each year. During the whole reanalysis period, about 13,000,000 observation data points were available.

### 2.3.2 Evaluation dataset: the AERONET AOD

We employed the AERONET AOP product (version 2, Level 2.0 (quality-assured)) for the independent evaluation of the reanalysis. Various studies of AOP retrieval and aerosol simulation and assimilation have used AERONET AOPs for

validation (Kinne et al., 2006; Levy et al., 2013; Sessions et al., 2015; Lynch et al., 2016; Rubin et al., 2016). Because the AERONET AOP product does not include AODs measured at 550 nm, we used the Ångström law to derive AODs at 550 nm from the AODs and Ångström exponents measured at multiple wavelengths (340–870 nm). Then the AERONET AODs were averaged into 1-hour bins and paired with the model results. We used observation data from 277 AERONET sites situated between 60°S and 60°N from which observations covering more than 5% of the reanalysis period were available

(Fig. 3). Roughly speaking, we obtained observation data from about 190 sites each month, but the number of available data varied seasonally and reached a maximum in boreal summer (Fig. 4). In the most recent year (i.e., 2015), the numbers of both available stations and data were smaller, because it takes time for Level 2.0 (quality-assured) data to be established; the Level 2.0 data become available only after final calibration has been applied and manual data inspection has been completed. To compare the modeled AODs with the AERONET AODs, the gridded modeled values were linearly interpolated to the

AERONET site locations.

### 2.4 Experiment setup

The configuration of the reanalysis is summarized in Table 1. The model resolution was set to TL159 (about 1.1° × 1.1°; 320 × 160 grid points) with 48 vertical layers from the ground to 0.4 hPa using the hybrid sigma pressure coordinate system. At the three lowest levels, the vertical resolution was about 100, 300, and 600 m. The time step of the aerosol model was 900

30  s, and we used hourly model output data for the evaluation. In this study, the operational global analysis provided by JMA (GANAL/JMA; JMA, 2002) at 6-hour intervals was used for nudging the AGCM. We used anthropogenic and biomass burning emissions of sulfur dioxide, BC, and OC from the MACCity (MACC/CityZEN EU projects) emission inventory





(http://ether.ipsl.jussieu.fr/eccad) and the Global Fire Assimilation System (GFAS) dataset (http://www.gmes-atmosphere.eu/about/project_structure/input_data/d_fire). This MASINGAR mk-2 setup, excepting the model resolution and the assimilation setting, is identical to the operational forecasting setup of the ICAP MME.

Figure 5 shows a schematic diagram of the reanalysis procedure. For this study, we performed two runs: a model simulation without data assimilation (free run; hereafter FR), and the reanalysis run (RA), in which NRL-UND MODIS AODs were assimilated every 6 hours. The reanalysis period was from 1 January 2011 to 31 December 2015 (5 years) with a spin-up period of three months (October–December 2010). We also compared the 6-hour forecast AODs from the analyzed state (first guess, hereafter FG) with the NRL-UND MODIS AODs.

## 3 Evaluation results

### 3.1 Chi-square test

We used the chi-square test ($\chi^2$), which evaluates the balance between the innovation and the background and observation error covariances, to evaluate the long-term stability of the assimilation performance (Ménard et al., 2000; Miyazaki et al., 2012, 2015). In this study, the chi-square value was defined as follows:

$$\mathbf{Y} = \frac{1}{\sqrt{m}}\left(\mathbf{R}+\mathbf{HP}_\tau\ \mathbf{H}_I^{\ T}\right)^{-1/2}(\boldsymbol{\tau}^o - \mathbf{H}_I\boldsymbol{\tau}^f), \qquad (27)$$

$$\chi^2 = \mathrm{trace}\mathbf{YY}^T. \qquad (28)$$

Here, $m$ is the number of observation data. When the background and observation error covariances properly balance the innovation, $\chi^2$ has the ideal value of 1. $\chi^2 > 1$ indicates overconfidence (i.e., underestimation) of the model or observation errors.

The time evolution of $\chi^2$ during RA (Fig. 2b) shows that it decreased in the first two months of the reanalysis period and then remained approximately constant with an average value of 0.30 and a standard deviation of 0.077. This result confirms that the assimilation performance was stable. The relatively large $\chi^2$ value in October 2013 was caused by the lack of assimilation data at that time. In almost all cases, $\chi^2$ was less than the ideal value of 1, which implies persistent overestimation of the background or observation errors with respect to the innovation. We performed an additional 2-year assimilation experiment (2011–2012) in which the background error covariances were uniformly decreased by 60%, and found that average $\chi^2$ value was 0.56 with a standard deviation of 0.16. Although the average was increased in this experiment, almost all of the $\chi^2$ values were still lower than the ideal value, which implies persistent overestimation of the observation errors.

### 3.2 Validation by MODIS AODs

To validate the data assimilation system, we compared the AOD fields from FR, FG, and RA with the NRL-UND MODIS AODs. Better agreement of RA AODs, compared with FR AODs, confirms the accuracy of the data assimilation system,



because NRL-UND MODIS AODs were used as an observational constraint. The FG performance is an indication of whether the analyzed aerosol fields can improve the short-term forecast.

The RA AOD spatial distribution showed quite good agreement with the NRL-UND MODIS AOD distribution (Figs. 6a and b). The distribution of the 5-year averaged increment (RA AOD minus FR AOD) (Fig. 6c) shows that, in general, assimilation increased AODs over the oceans, implying that in FR, MASINGAR mk-2 underestimated sea-salt aerosols. Large negative and positive increments over Canada, Siberia, and Indonesia resulted from improved simulation of carbonaceous aerosols from forest fires. Over the Sahel (south of the Sahara Desert) and its downwind regions (e.g., the Atlantic Ocean), mineral dust particles were increased by assimilation. In contrast, over other dust-dominant regions (central China, Australia, Persian Gulf, and Argentina), AODs (mainly mineral dust particles) were decreased. The negative increment around the Mediterranean Sea resulted from decreased Saharan dust in RA. Over industrializing areas, such as India and the eastern coast of China, assimilation increased anthropogenic pollutants (i.e., sulfate and carbonaceous aerosols), filling the gap between observed and FR AODs. The large positive increments around central Africa and the Gulf of Guinea reflect increased carbonaceous aerosols due to forest fires.

Temporal evolution of the root mean square error (RMSE), linear Pearson correlation coefficient ($R$), mean fractional error (MFE), mean fractional bias (MFB), and index of agreement (IOA) for FR, FG, and RA AODs are shown in Fig. 7 (formulations of these statistical measures are given in Appendix A). For FR, these statistical measures (blue lines and dots) showed seasonal variations reflecting the seasonal cycle of the observation coverage (shown in Fig. 1) and the aerosol distribution (e.g., springtime Asian dust storms). Occasional large RMSEs and low $R$ values for FR AODs (e.g., July 2011 and March 2015) were caused mainly by large-scale dust storms or forest fires. Values of the RA AOD statistical measures (red lines and dots) are much better than the FR values. In most cases, the RMSEs are less than 0.06 (the 80th percentile of RMSE for RA is 0.057), and $R$ and IOA values are larger than 0.9 (20th percentile of $R$ and IOA for RA is 0.91 and 0.94, respectively). The 80th percentile of MFE for RA AOD is 26.6%. The MFB time series showed that assimilation considerably reduced the bias found in FR. During 96.4% of the reanalysis period, MFB for RA AODs is within ±10%. In most cases, RA AODs meet the model performance goals for particulate matter and light extinction (MFE ≤ +50% and MFB ≤ ±30%) suggested by Boylan and Russell (2006). Moreover, all FG AOD (green lines and dots) statistical measures are better than the FR AOD statistics. This result means that aerosol fields obtained by the reanalysis improved short-term (6–24 hour) forecasting.

Scatter plots of FR, FG, and RA AODs versus NRL-UND MODIS AODs for the whole reanalysis period (Fig. 8) and by season (Fig. 9) show no seasonal dependency of RA. However, MFB values are relatively larger in boreal spring and summer, perhaps reflecting larger AOD values over the Northern Hemisphere during those seasons (Table 2). RA AODs are much more aligned with the 1:1 line in the scatter plots compared to both FR and FG AODs, and the RA statistical measures are much better, both seasonally and for the whole analysis period (RMSE ≤ 0.05, $R$ ≥ 0.95, MFE ≤ 25.1%, MFB = 1.3–4.1%, and IOA ≥ 0.97; Table 2). The scatter diagrams show that RA AODs tended to be underestimated with respect to NRL-UND MODIS AODs. In fact, the mean bias (averaged value of RA AODs minus NRL-UND MODIS AODs) is



slightly negative (–0.007). However, MFB is slightly positive (2.8%). This discrepancy implies that although RA underestimated relatively large AODs (>0.5), smaller AODs (<0.5) were overestimated. The FG AODs also show better agreement than FR AODs with the NRL-UND MODIS AODs throughout the reanalysis period (Table 2) and meet the model performance goal. The frequency distribution of FR AOD deviations (observed AODs minus modeled AODs) (Fig. 10) is

5 symmetric and mound-shaped, similar to a Gaussian distribution, and supports the assumption of the data assimilation that the background errors follow a Gaussian distribution. However, the requirement of an unbiased background error is not strictly met (mean bias between FR AODs and NRL-UND MODIS AODs is –0.032). FG and RA AOD deviations shows squeezed distributions, and their peaks are closer to 0 than the FR peak; 92.0% (79.6%) of RA (FG) AOD deviations are within ±0.05.

## 3.3 Evaluation by AERONET AODs

### 3.3.1 Evaluation with 1-hour binned data

We compared the modeled AODs with the 1-hour binned AERONET AODs during the whole reanalysis period (Fig. 11) and during each season (Fig. 12) in scatter plots. In general, the FR AOD distribution is squashed vertically, an indication that FR AODs generally underestimated AERONET AODs. This underestimation is also reflected in negative MFB values

(Table 3). In boreal spring (Fig. 12c) and summer (Fig. 12e), FR AODs in regions where AERONET AODs were between 0.0 and 1.0 showed positive biases, which were caused by the overestimation of carbonaceous aerosols from forest fires in Canada and mineral dust aerosols over central China, Australia, and the Persian Gulf mentioned in Sect. 3.2. RA AODs (Fig. 11b; Figs 12b, d, f, and g) are more narrowly distributed along the 1:1 line, an indication that assimilation improved under- and overestimates in the FR AODs. The statistical measures of RA for the whole reanalysis period (Table 3) meet the model

performance goals. The MFE and MFB values obtained in boreal summer were better compared with those obtained in other seasons. This improvement might reflect the relatively larger number of NRL-UND MODIS AODs available in boreal summer (see Table 2).

The distribution of FR AOD deviations (AERONET AODs minus modeled AODs) (Fig. 13, blue line) is positively biased (i.e., FR AODs underestimate AERONET AODs); 70.0% of the deviations are larger than 0 (2.1% exceed 0.5). The

25 distribution of RA AOD deviations (Fig. 13, red line) shows less positive bias and is more symmetric compared with the FR AOD result; 48.4% of the deviations are larger than 0 (1.1% exceed 0.5). Although the distribution of RA AOD deviations relative to AERONET AODs is broader, compared with the distribution relative to NRL-UND MODIS AODs (Fig. 10), 73.1% (80.9%) of the deviations are still within ±0.05 (±0.10).

### 3.3.2 Evaluation with monthly averaged data

We also evaluated monthly averaged AODs, because monthly AOD data are often used in climate studies (Lynch et al., 2016). Monthly averaged AODs were derived from paired 1-hour binned AERONET and modeled AODs. In general, the



monthly averaged RA AODs (Table 4) showed better agreement than the 1-hour binned AODs (Table 3) with the AERONET AOD data; in particular, RMSE and MFB of RA were about 43% and 86% lower, respectively. As in the pairwise comparison with the 1-hour binned AODs, FR AODs underestimated the monthly averaged AERONET AODs (Fig. 14a and negative MFB value in Table 4), whereas RA AODs reduced or eliminated the underestimation. As a result, the RA

MFB value, 0.6%, is quite good (Table 4). Moreover, in the monthly data, 74.0% (89.4%) of RA AOD deviations from AERONET AODs were within ±0.05 (±0.10).

Time series of the statistical measures calculated by using the monthly averaged AERONET AODs (Fig. 15) show that, in general, RA AOD performance is much better than FR AOD performance throughout the reanalysis period, except for MFB in December 2016. The only statistical measure that shows seasonality is MFE. RMSE of RA is almost always (58 months

during 5 years) lower than 0.10; 53.3% (32 months) of $R$ values and 73.3% (44 months) of IOA values are greater than 0.90; 80.0% (48 months) of MFE and MFB values are less than 34% and within ±10%, respectively; and all months except for the initial month of the reanalysis (i.e., January 2011) meet the model performance goals.

Statistical measures for RA at each AERONET site are shown in Fig. 16, and those for FR AOD are shown in Fig. 17 as a reference. In these figures, the 181 AERONET sites with more than 36 monthly averaged data are plotted. In general, RA

shows much better agreement globally than FR with the AERONET data. In RA, RMSE < 0.01, $R$ > 0.90, and IOA > 0.90 at 86.4% (154 sites), 40.7% (74 sites), and 43.4% (79 sites) of the 181 sites, respectively. MFE is less than 50% at 181 sites (91.2%), and MFB values at 81 sites (44.5%) are within ±15%. Consequently, at 44.5% (81 sites) of the 181 AERONET sites, RA meets the model performance goal. RMSEs are relatively large at sites in central Africa, India, Southeast Asia, and eastern coastal China, but the other statistical measures (e.g., $R$ and MFE) in these regions are not particularly worse than

those in the other regions. Relatively higher AOD values (see Fig. 6) are responsible for the larger RMSE values.

At Beijing (39.977°N, 116.381°E; Fig. 18b), RMSE (0.28) and the negative bias (MFB = –49.3%) are large, and IOA is relatively low (0.65), but the correlation coefficient is relatively high ($R$ = 0.80). We also compared time series of monthly averaged AERONET and modeled AODs with the time series of monthly averaged NRL-UND MODIS AODs there in Fig. 18. It should be noted that the monthly averaged MODIS AODs are derived by using all available data, and neither the

25 AERONET nor the modeled AODs are paired. At Beijing, although RA successfully captured the temporal variation observed by AERONET, its AOD values were almost always lower than the observed values. A similar negative bias when AOD levels are high has been observed for other megacities in industrializing countries, including XianHe (39.754°N, 116.962°E; about 56 km from Beijing), Kanpur, India (26.513°N, 80.232°E; Fig. 18h), and Mexico_City, Mexico (19.334°N, 99.182°W). The statistics for these sites are summarized in Table 5. Averaged RA AODs are lower than averaged

AERONET AODs, and MFB values are negative (–22.0% to –53.7%), but correlation coefficients are relatively high ($R$ = 0.78–0.79). Multi-model inter-comparison studies (Kinne et al., 2006; Sessions et al., 2015) have pointed out that aerosol models having negative biases for high-AOD events is a common problem. Insufficient anthropogenic emissions data and model resolution for megacities are plausible reasons for the negative biases. Assimilation partly reduced the negative bias (see Figs. 18b and 18h), although the amount of improvement was limited. The probability of a successful retrieval can be



reduced during high-AOD events (Lynch et al., 2016); thus, fewer available satellite observations over megacities during high-AOD events may also account for the negative biases in RA AODs.

At Lulin (23.469°N, 120.874°E; Fig. 18i), the modeled AODs showed a positive bias (MFE and MFB of RA AODs are 135.6% and 135.5%, respectively), and the improvement due to assimilation was limited. We found similar positive biases

with little improvement at QOMS_COM, China (28.365°N, 86.948°E), CASLEO, Argentina (31.799°S, 69.306°W), and Mauna_Loa, Hawaii (19.539°N, 155.578°W) (Table 5). All of these sites have large MFE and MFB values; when MFE is equal to MFB, the modeled AODs are always larger than the observed AODs. One characteristic shared by these sites is a relatively low AOD level (0.025–0.056), and another shared feature is that all four sites are situated at high elevation in a mountainous area (Table 5). The coarse model resolution might obscure the effect of local terrain and result in this positive

bias. In addition, the NRL-UND MODIS AODs also overestimated the AERONET AODs because of the difference in the representativeness of the observations. This fact can explain the limited improvement by the assimilation.

In addition to the sites discussed above, we show time series of monthly averaged AERONET and modeled AODs of other sites (Fig. 18), which we selected as representative of urban areas (GSFC, Moldova, and Osaka) or because they show the influence of African dust (IER_Cinzana, Capo_Verde, and Ragged_Point), Arabian dust (Mezaira), African smoke

(Ascension_Island), Southeast Asian smoke (Chiang_Mai_Met_Sta), or South American smoke (Rio_Branco). Cart_Site was selected as representative of a clean area. Nine of these sites were also selected for validation of the ICAP-MME (Sessions et al., 2015). At the urban sites (Figs. 18f, k, and i), assimilation improved the negative bias in FR AODs, and RA AODs showed much better agreement with the observations than FR AODs, in particular at GSFC (Fig. 18f), although a slight negative bias (MFB = –1.9%) remained in RA AODs. At IER_Cinzana (Fig. 18g), situated in an African dust source

region, RA AODs successfully captured the observed temporal variation ($R$ = 0.93, IOA = 0.89), but a negative bias (MFB = –32.7%) remained after the assimilation. The availability of fewer NRL-UND MODIS AOD observations over bright surfaces (also see Fig. 1) can explain the limited improvement achieved by assimilation. At Capo_Verde (Fig. 18c) and Ragged_Point (Fig. 18m), which are downwind of an African dust source, the RA results showed nearly perfect agreement with observations, though the sites are on opposite sides of the Atlantic Ocean. At Mezaira (Fig. 18j), which is influenced by

Arabian dust, AERONET captured the AOD peaks in boreal summer. FR considerably overestimated the AOD values of the peaks, and assimilation improved the overestimation, even though no MODIS data are available over this AERONET site. MODIS observations near the site (i.e., over the Persian Gulf; see Fig. 1) contributed to this improvement, and the overall performance of RA was good (Fig. 18j). At Ascension_Island (Fig. 18g), which is influenced by African smoke, assimilation improved the persistent negative bias. In RA AODs, RMSE = 0.03 and $R$ = 0.92, and the negative bias (MFB = –3.0%) was

much smaller than that in FR AODs. At Chiang_Mai_Met_Sta, which is influenced by Southeast Asian smoke (Fig. 18e), RA AODs reproduced the seasonal variation (peaks in spring) seen in the AERONET AODs ($R$ = 0.96), but they underestimated peak AOD values (MFB = –12.9%). The NRL-UND MODIS peak AOD values were also lower than the AERONET peak AODs. This fact can explain the negative bias of the peak AOD values in the RA AODs. At Rio_Branco (Fig. 18n), which is influenced by South American smoke, RA AODs successfully captured both the temporal variation and



the winter peaks ($R = 0.93$), but peak AOD values were slightly underestimated (MFB = –1.5%). At the clean site (Cart_Site; Fig. 18d), assimilation complemented the underestimates in FR AODs, and the RA performance was good (RMSE = 0.02, $R$ = 0.94, MFE = 15.3%, MFB = –2.5%, and IOA = 0.96).

**4 Conclusions**

A global aerosol data assimilation system was developed based on a global aerosol transport model, MASINGAR mk-2, and a 2-dimensional variational data assimilation method. The assimilation system was used to produce an aerosol reanalysis product named the Japanese Reanalysis for Aerosol (JRAero) for the period 2011–2015 through the assimilation of NRL-UND MODIS AOD data with a horizontal resolution of TL159 (about 1.1° × 1.1°). In this paper, we have outlined the data

assimilation system and presented the general specifications of JRAero. We have also presented results of our validation of the reanalysis AODs with AERONET AODs.

Chi-square test results confirmed that the stability of the assimilation performance was good throughout the reanalysis period. In almost all cases, the chi-square value was lower than the ideal value of 1; this result implies persistent overestimation of the background and observation errors with respect to the innovation.

Comparison of the reanalysis results with the NRL-UND MODIS AODs showed that assimilation improved both negative and positive biases in the free run (FR) of MASINGAR mk-2: Negative biases over oceans caused by underestimation of sea-salt aerosols, gaps in carbonaceous aerosols from forest fires in Canada, Siberia, Indonesia, and central Africa, discrepancies in dust source and downwind regions (e.g., Sahel, Atlantic Ocean, Mediterranean Sea, central China, Australia, and Persian Gulf), and underestimates in industrializing areas (in particular, India and eastern China) were compensated by

the assimilation. The reanalysis AODs showed quite good agreement with the NRL-UND MODIS AODs (RMSE = 0.05, $R$ = 0.96, MFE = 23.7%, MFB = 2.8%, and IOA = 0.98), confirming the accuracy of the assimilation system. FG AODs also showed better agreement with the NRL-UND MODIS AODs than the FR AODs (although worse compared with the reanalysis AODs). This result indicates that aerosol fields provided by the reanalysis are capable of substantially improving short-term (6–24 hour) forecasting.

Validation with 1-hour binned AERONET AODs showed the reanalysis AODs to be considerably better than the FR AODs. The statistical measures of RA (RMSE = 0.14, R = 0.76, MFE = 42.7%, MFB = 4.5%, and IOA = 0.85, based on about 1,500,000 1-hour binned AOD data from 277 AERONET sites) met the model performance goals proposed by Boylan and Russell (2006). In addition, 73.1% (80.9%) of deviations (the AERONET AOD minus the reanalysis AOD) were within ±0.05 (±0.10).

The statistics of the reanalysis AODs in a comparison with monthly averaged AERONET AODs (RMSE = 0.08, R = 0.90, MFE = 28.1%, MFB = 0.6% and IOA = 0.93) were even better than those in the comparison with 1-hour binned AERONET AODs. In site-by-site comparisons, the reanalysis performance was also better than the FR performance. At 86.4% of the 181 AERONET sites, the RMSE of RA was <0.10; at 40.7% of sites, $R$ was >0.90; and at 43.4% of sites, IOA was >0.90.



However, the reanalysis AODs tended to underestimate the observed AODs at urban sites (in particular, megacities in industrializing countries), possibly because anthropogenic emissions data and model resolution were insufficient. At high-elevation mountain sites (2500–4200 m), persistent positive biases were found in the reanalysis and improvement by assimilation was limited. The coarse model resolution, which likely obscures the effect of local terrain, and the difference in

representativeness between satellite and ground-based observations can explain the overestimation and limited improvement.

## 5 Future directions

To enhance the current version's quality and address its problems, we propose the following for the next version:

  1. NRL-UND MODIS AODs (or Dark Target AODs) are unavailable over deserts. These regions without observational constraints are responsible for the limited improvement of RA obtained near dust source regions. Inclusion of AOD data

retrieved by the "Deep Blue" algorithm, which is able to complement Dark Target AODs by retrieving AODs over bright arid land surfaces (Hsu et al., 2006; Sayer et al., 2013), could improve the reanalysis quality in regions around deserts. In the current version, we assimilated 2-dimensional maps of AODs (vertically integrated aerosol optical property) and assumed that the shape of vertical profile before assimilation was unchanged after assimilation. However, vertical profiles of aerosols likely affect their transport, deposition, and climate effects. A space-borne lidar, CALIPSO (Winker et al., 2010), has been

providing continuous measurements of aerosol vertical distributions over the globe since 2006. The use of lidar data would allow the vertical profiles of the reanalysis to be adjusted. Furthermore, the inclusion of information on the size distribution (e.g., Ångström exponent) would raise the quality of the reanalysis.

  2. At megacity and mountain sites, assimilation provided limited improvement, and positive and negative biases remained in the reanalysis. A plausible reason is the coarse model resolution, which is insufficient to resolve high-AOD events around

megacities and local terrain effects in mountainous areas. Therefore, we plan to rerun the reanalysis with a finer resolution and check the performance of the model.

  3. In the next version, we will employ the JRA-55 reanalysis dataset (Kobayashi et al., 2015; Harada et al., 2016), the most recent meteorological reanalysis product developed by JMA, as nudging data in the AGCM. The use of these data is expected to enable more accurate simulation of aerosol transport and deposition and to allow longer integration of the

reanalysis.

  4. Updating to more recent anthropogenic emissions inventory data will improve the negative biases, especially for industrializing countries where anthropogenic emissions are rapidly increasing.

  5. In addition to the updates of the model resolution and input data, physical and chemical processes in the model need to be sophisticated. Newly developed dust emission (e.g., Kang et al., 2014), optical calculation, microphysics parameterization,

and wet deposition schemes (e.g., Oshima et al, 2009; Oshima and Koike, 2013) will be applied in the next version. Aerosol microphysical and optical parameters (e.g., size distribution, hygroscopicity, and refractive index) will also be updated by taking advantage of recent measurements.





6. The chi-square test results imply model and observation errors relatively large with respect to the innovation. Re-examination of these errors is required. Insights from studies using an ensemble-based assimilation system (Yumimoto et al., 2016) should make it possible to provide the background error covariance matrix with a more appropriate amplitude and structure.

7. Further validation of the reanalysis is also needed. Comparison with vertical AOD distributions obtained from space- and ground-based lidars would contribute to improvement of the vertical profiles of the reanalysis and become a good prearrangement of assimilation with the lidar data. Validation using mass concentrations observed in situ and by aircraft campaigns should effectively improve the model's emission, transport, diffusion, and deposition processes. Furthermore, inter-comparison with other reanalysis products (e.g., CAMSiRA, MERRA-2, and NRL NAAPS reanalyses) will provide insight into various aspects of not only the data assimilation system but also the aerosol model and help us increase their sophistication.

**Code and data availability**

The model and data assimilation system codes are the property of MRI/JMA and not available to the general public. If you are interested in the codes, please contact the corresponding author. The reanalysis product is available from the authors upon request.

**Appendix A: Statistical metrics**

We used the root mean square error (RMSE), the (Pearson) correlation (R), the mean fractional error (MFE), the mean fractional bias (MFB; also known as the fractional gross error), and the index of agreement (IOA) for validating the reanalysis. The statistical metrics are defined as follows:

$$\text{RMSE} = \sqrt{\frac{1}{N}\sum_{i=1}^{N}(M_i - O_i)^2} \ , \tag{A1}$$

$$\text{R} = \frac{\sum_{i=1}^{N}(O_i - \bar{O})(M_i - \bar{M})}{\sqrt{\sum_{i=1}^{N}(O_i - \bar{O})^2 \sum_{i=1}^{N}(M_i - \bar{O})^2}} \ , \tag{A2}$$

$$\text{MFE} = \frac{2}{N}\sum_{i=1}^{N}\frac{|M_i - O_i|}{M_i + O_i}\times 100 \ , \tag{A3}$$

$$\text{MFB} = \frac{2}{N}\sum_{i=1}^{N}\frac{M_i - O_i}{M_i + O_i}\times 100 \ , \tag{A4}$$

$$\text{IOA} = 1 - \frac{\sum_{i=1}^{N}(O_i - M_i)^2}{\sum_{i=1}^{N}(|O_i - \bar{O}| + |M_i - \bar{M}|)^2} \ , \tag{A5}$$



$N$ is the total number of pairs of modeled ($M$) and observed ($O$) values. $\bar{O}$ and $\bar{M}$ represent $\frac{1}{N}\sum_{i=1}^{N} O_i$ and $\frac{1}{N}\sum_{i=1}^{N} M_i$, respectively. RMSE represents the standard deviation of the discrepancies between modeled and observed values. MFE can range from 0 to 200%, and is a measure of overall modeling error without emphasizing outliers. MFE = 0 is a perfect score. MFB is a measure of the estimation bias error that allows symmetric analysis of over- or underestimation by the model relative to observed values. The best value of MFB is 0 with ±200% the minimum and maximum values. Boylan and Russell (2006) proposed that MFE should be ≤ +50% and MFB should be ≤ ±30% to meet model performance goals for particulate matter (PM) and light extinction. IOA was developed by Willmott (1981) as a standard measure of the degree of model prediction accuracy, and it ranges from 0 to 1. IOA = 1 indicates perfect agreement.

**Author contribution**

K. Yumimoto developed the assimilation system, designed the reanalysis, and performed the validation. T.Y. Tanaka developed and maintained the aerosol transport model. N. Oshima updated the calculation of aerosol optical depth. T. Maki contributed to the optimization of the system. K. Yumimoto wrote the initial draft of manuscript, and all authors carefully reviewed the manuscript.

**Acknowledgements**

This work was supported by the Japan Society for the Promotion of Science (JSPS) KAKENHI Grant Numbers JP25220101, JP26701004, JP15K05296, and JP16H02946, and by the Environment Research and Technology Development Fund (No. S-12) and the Global Environment Research Fund (2-1403) of the Japanese Ministry of Environment. We thank all of the Principal Investigators and all those who have contributed to the establishment and maintenance of the AERONET sites, the NRL-UND MODIS L3 AOD product, and the NASA MODIS L2 AOD products. Staff members of the Atmospheric Environment and Applied Meteorology Research Division of MRI are acknowledged for their advice and comments.

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



**Table 1: Configuration of the reanalysis.**

| | |
|---|---|
| Aerosol model | MASINGAR mk-2 |
| Meteorological model | MRI-AGCM3 nudging with GANAL/GANAL |
| Model resolution | TL159 (1.1° × 1.1°) with 48 vertical layers |
| Assimilation method | 2-dimensional variational method |
| Assimilation data | NRL-UND MODIS AOD (L3) |
| Assimilation interval | 6 hours |
| Localization scale | 1000 km |
| Product period | 2011–2015 (5 years) |



**Table 2: Statistical measures of the NRL-UND MODIS AODs versus FR, FG, and RA AODs.**

| | RMSE | | | *R* | | | MFE | | | MFB | | | IOA | | | # of data |
|---|---|---|---|---|---|---|---|---|---|---|---|---|---|---|---|---|
| | FR | FG | RA | FR | FG | RA | FR | FG | RA | FR | FG | RA | FR | FG | RA | |
| Whole period | 0.14 | 0.11 | 0.05 | 0.61 | 0.78 | 0.96 | 56.8 | 39.7 | 23.7 | –24.3 | 6.3 | 2.8 | 0.76 | 0.88 | 0.98 | 12960929 |
| DJF | 0.14 | 0.11 | 0.05 | 0.60 | 0.77 | 0.96 | 61.4 | 39.4 | 23.3 | –33.3 | 4.0 | 1.3 | 0.75 | 0.87 | 0.97 | 2774132 |
| MAM | 0.15 | 0.11 | 0.05 | 0.64 | 0.79 | 0.96 | 56.8 | 38.9 | 23.4 | –26.0 | 7.2 | 3.4 | 0.78 | 0.89 | 0.98 | 3182020 |
| JJA | 0.16 | 0.12 | 0.05 | 0.61 | 0.79 | 0.96 | 54.0 | 41.4 | 25.1 | –14.8 | 8.3 | 4.1 | 0.75 | 0.88 | 0.98 | 3811923 |
| SON | 0.13 | 0.10 | 0.05 | 0.58 | 0.76 | 0.95 | 56.3 | 38.9 | 22.7 | –26.0 | 5.1 | 1.9 | 0.72 | 0.87 | 0.97 | 3192854 |

RMSE, root mean square error; *R*, correlation coefficient; MFE, mean fractional error; MFB, mean fractional bias; IOA, index of agreement.

FR, free run; FG, first guess; RA, reanalysis.

DJF, boreal winter (December–February); MAM, boreal spring (March–May); JJA, boreal summer (June–August); SON, boreal autumn (September–November).

**Table 3: Statistical measures of the 1-hour binned AERONET AODs versus FR and RA AODs.**

| | RMSE | | *R* | | MFE | | MFB | | IOA | | # of data |
|---|---|---|---|---|---|---|---|---|---|---|---|
| | FR | RA | FR | RA | FR | RA | FR | RA | FR | RA | |
| Whole period | 0.19 | 0.14 | 0.58 | 0.76 | 56.2 | 42.7 | –27.2 | 4.5 | 0.71 | 0.85 | 1513663 |
| DJF | 0.19 | 0.14 | 0.58 | 0.77 | 65.6 | 50.3 | –31.9 | 5.6 | 0.69 | 0.86 | 254833 |
| MAM | 0.21 | 0.16 | 0.59 | 0.77 | 56.8 | 41.5 | –27.5 | 6.7 | 0.73 | 0.86 | 411600 |
| JJA | 0.20 | 0.15 | 0.53 | 0.74 | 51.7 | 38.8 | –27.0 | 2.3 | 0.69 | 0.84 | 497808 |
| SON | 0.16 | 0.13 | 0.59 | 0.76 | 55.0 | 44.1 | –23.8 | 4.4 | 0.69 | 0.84 | 349422 |

**Table 4: Statistical measures of the monthly averaged AERONET AODs versus FR and RA AODs.**

| | RMSE | | *R* | | MFE | | MFB | | IOA | | # of data |
|---|---|---|---|---|---|---|---|---|---|---|---|
| | FR | RA | FR | RA | FR | RA | FR | RA | FR | RA | |
| Whole period | 0.12 | 0.08 | 0.76 | 0.90 | 44.5 | 28.1 | –27.3 | 0.6 | 0.81 | 0.93 | 11222 |





**Table 5: Statistical measures of the monthly AERONET AODs versus FR and RA AOD at megacity and mountain sites.**

| Notation | Site | Location | Elevation (m) | Observation | RA | | | | | |
|---|---|---|---|---|---|---|---|---|---|---|
| | | | | Average | Average | RMSE | R | MFE | MFB | IOA |
| Megacity | Beijing | (39.977°N, 116.381°E) | 92 | 0.56 | 0.33 | 0.28 | 0.79 | 50.4 | –49.3 | 0.65 |
| | XiangHe | (39.754°N, 116.962°E) | 36 | 0.60 | 0.37 | 0.27 | 0.78 | 44.7 | –44.6 | 0.66 |
| | Kanpur | (26.513°N, 80.232°E) | 123 | 0.61 | 0.48 | 0.14 | 0.86 | 23.6 | –22.0 | 0.80 |
| | Mexico_City | (19.334°N, 99.182°W) | 2268 | 0.28 | 0.17 | 0.12 | 0.84 | 53.7 | –53.7 | 0.59 |
| Mountain | Lulin | (19.334°N, 99.182°W) | 2868 | 0.056 | 0.24 | 0.20 | 0.74 | 136.5 | 135.3 | 0.40 |
| | QOMS_COM | (28.365°N, 86.948°E) | 4276 | 0.041 | 0.16 | 0.14 | 0.61 | 119.8 | 119.8 | 0.25 |
| | CASLEO | (31.799°S, 69.306°W) | 2552 | 0.025 | 0.08 | 0.06 | 0.71 | 108.6 | 108.6 | 0.24 |
| | Mauna_Loa | (19.539°N, 155.578°W) | 3397 | 0.051 | 0.14 | 0.13 | 0.51 | 162.3 | 162.3 | 0.09 |







**Figure 1.** Horizontal distribution of the number of NRL-UND MODIS AOD data assimilated in RA in (a) the whole reanalysis
period (2011–2015), (b) December–February (DJF), (c) March–May (MAM), (d) June–August (JJA), and (e) September–
November (SON).





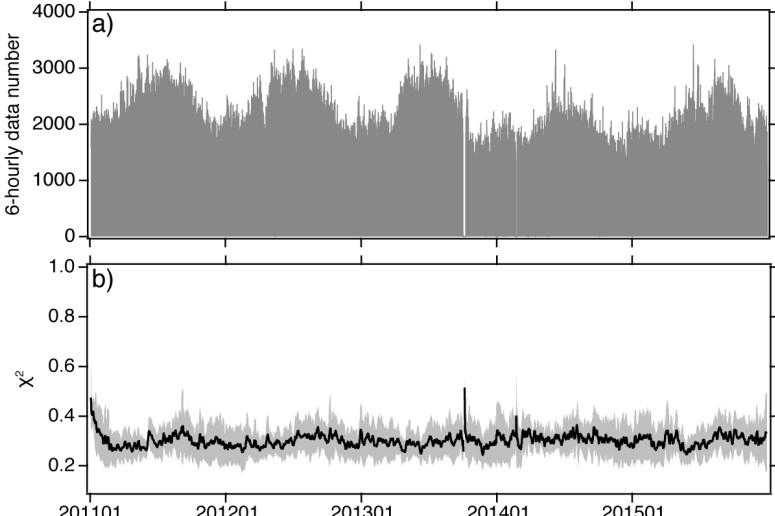

**Figure 2. (a) Time series of the number of 6-hourly NRL-UND MODIS AOD data. (b) Time series of the 5-day-moving average of the chi-square value ($\chi^2$; thick black line) and its standard deviation (gray shading).**





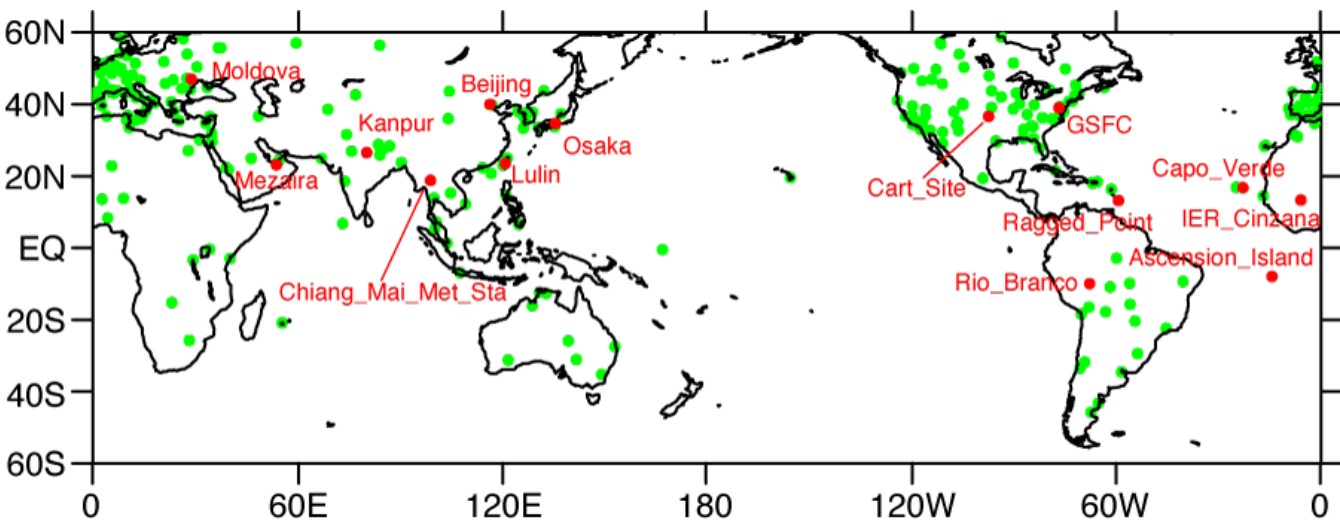

**Figure 3. AERONET sites used in this study. Red circles denote the AERONET sites used in Fig. 18.**





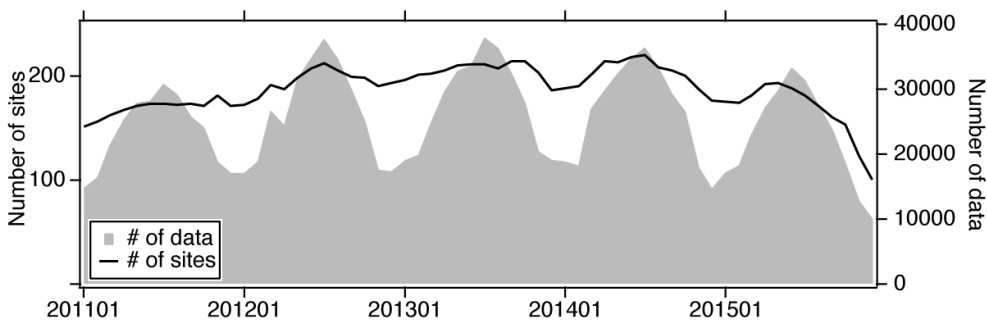

**Figure 4. Time series of the number of monthly AERONET AOD data (gray shading) and the number of AERONET sites (thick black line) used in the evaluation.**





**Figure 5. Schematic diagram of the reanalysis procedure. A map showing NRL-UND MODIS AODs is also shown at each assimilation time.**





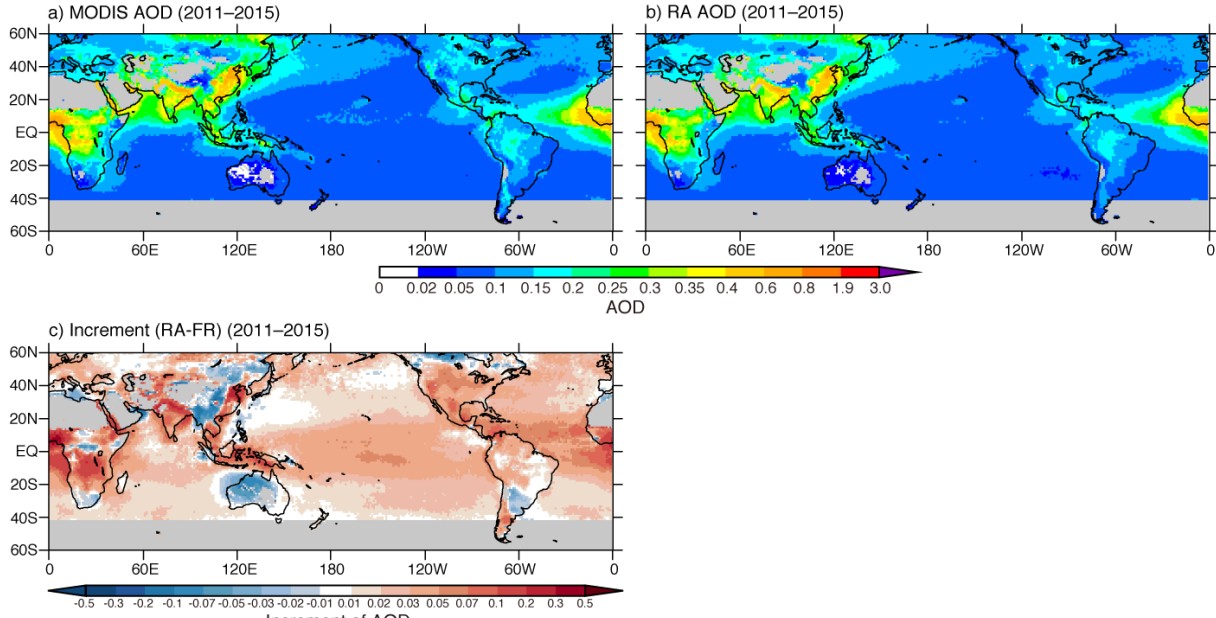

**Figure 6. Spatial distributions of 5-year averaged AODs during 2011–2015: (a) NRL-UND MODIS, (b) RA, and (c) increment between RA and FR. No observation data were available from gray areas (also see Fig. 1a).**





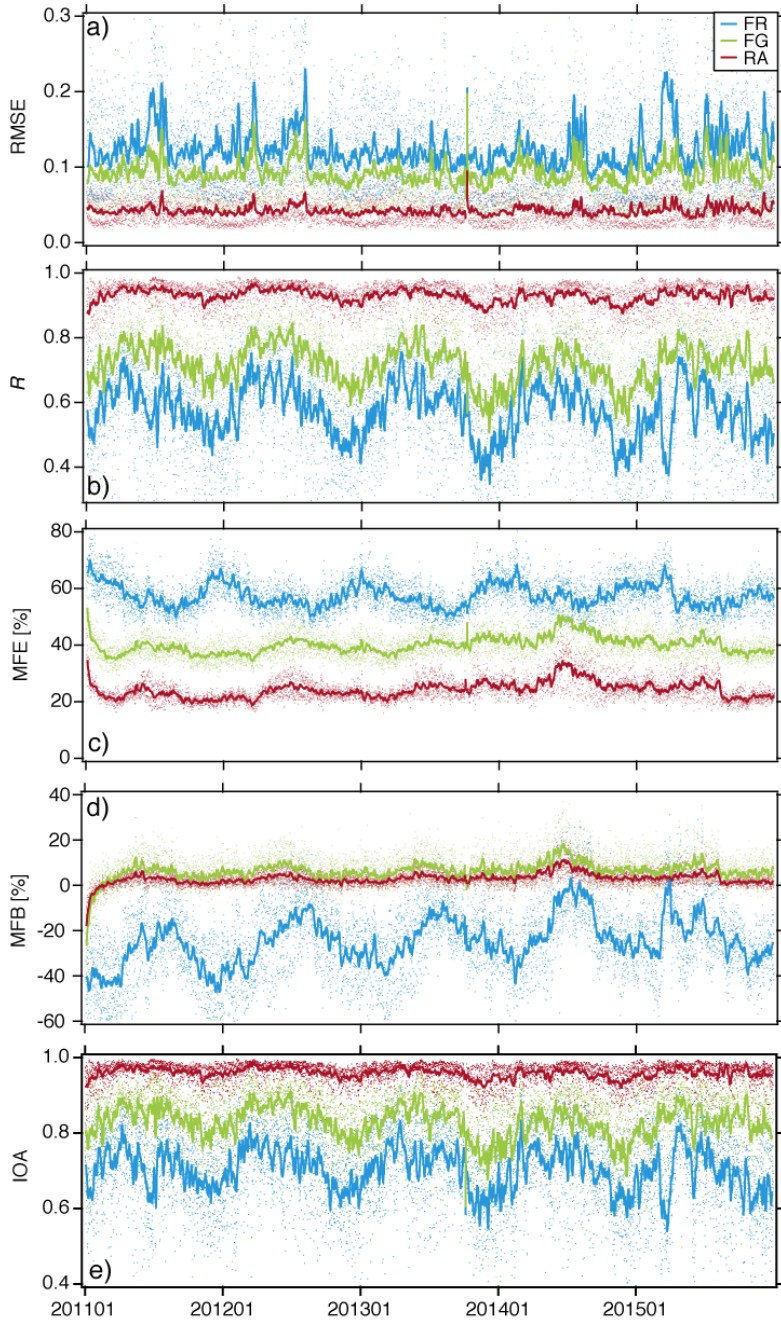

**Figure 7. Time series of the 5-day moving average of (a) RMSE, (b) *R*, (c) MFE, (d) MFB, and (e) IOA for FR (blue), FG (green), and RA (red), validated by NRL-UND MODIS AODs. Small dots show 6 hourly values.**





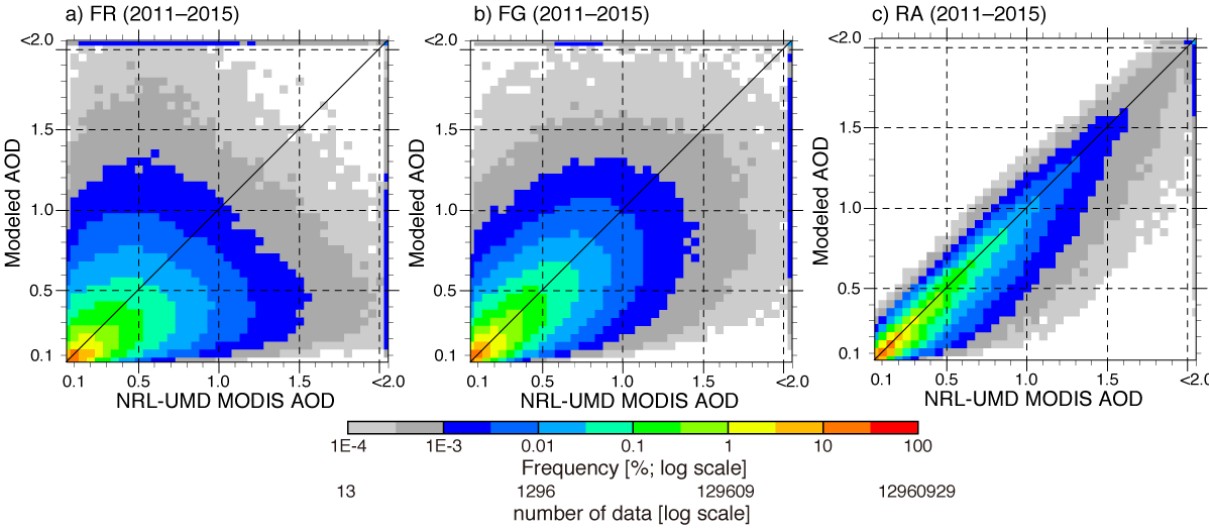

**Figure 8. Scatter plots of (a) FR, (b) FG and (c) RA AODs versus NRL-UND MODIS AODs during the whole reanalysis period (2011–2015). The AOD resolution is 0.05.**





**Figure 9. Scatter plots of FR (a, d, g, and j), FG (b, e, h, and k), and RA (c, f, i, and m) AODs versus NRL-UND MODIS AODs for boreal (a–c) winter (December–February), (d–f) spring (March–May), (g–i) summer (June–August), and (j–m) autumn (September–November).**





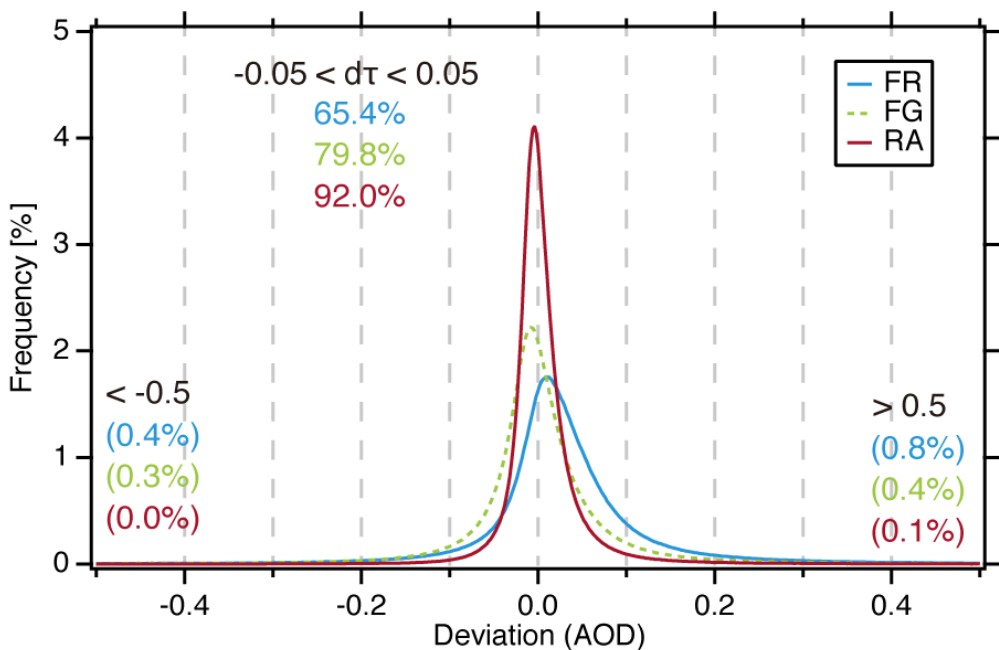

**Figure 10. Frequency distributions of deviations (observed AODs minus modeled AODs) from the NRL-UND MODIS AODs. The percentages of deviations between –0.05 and +0.05 are shown at the top, and the percentages less than –0.5 (bottom left) or greater than +0.5 (bottom right) are shown in parentheses.**





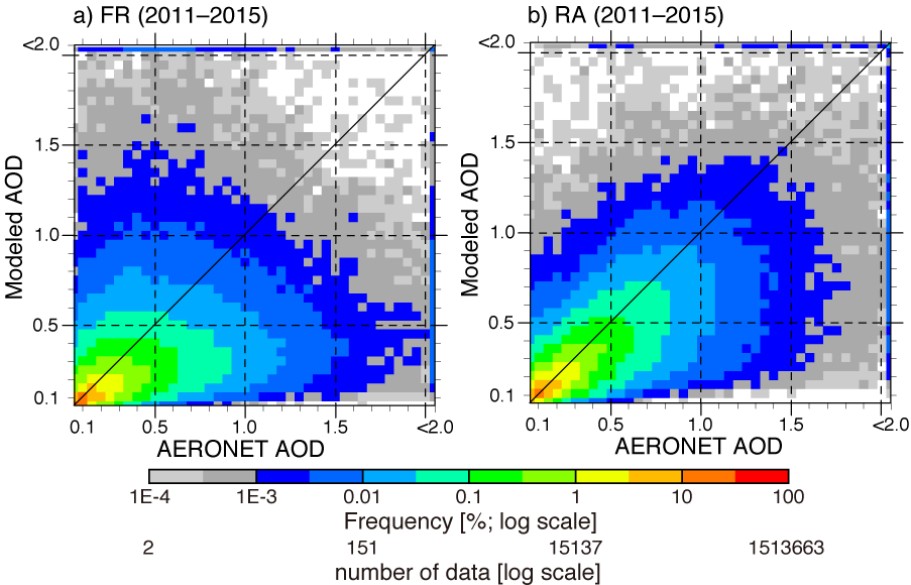

**Figure 11. Scatter plots of (a) FR and (b) RA AODs versus 1-hour binned AERONET AODs for the whole reanalysis period (2011–2015)**



**Figure 12. Same as Fig. 11, except for boreal (a–b) winter (December–February), (c–d) spring (March–May), (e–f) summer (June–August), and (g–h) autumn (September–November).**





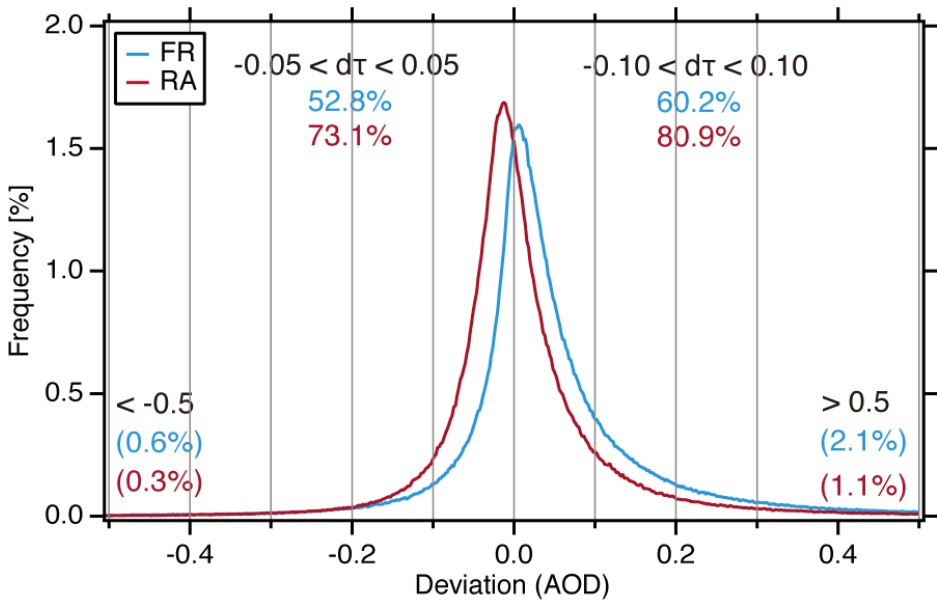

**Figure 13. Frequency of deviations (observed AODs minus modeled AODs) from the 1-hour binned AERONET AOD. The percentages of deviations between –0.05 and +0.05 (–0.10 and +0.10) are shown at the top left (top right), and the percentages less than –0.5 (bottom left) or greater than +0.5 (bottom right) are shown in parentheses.**



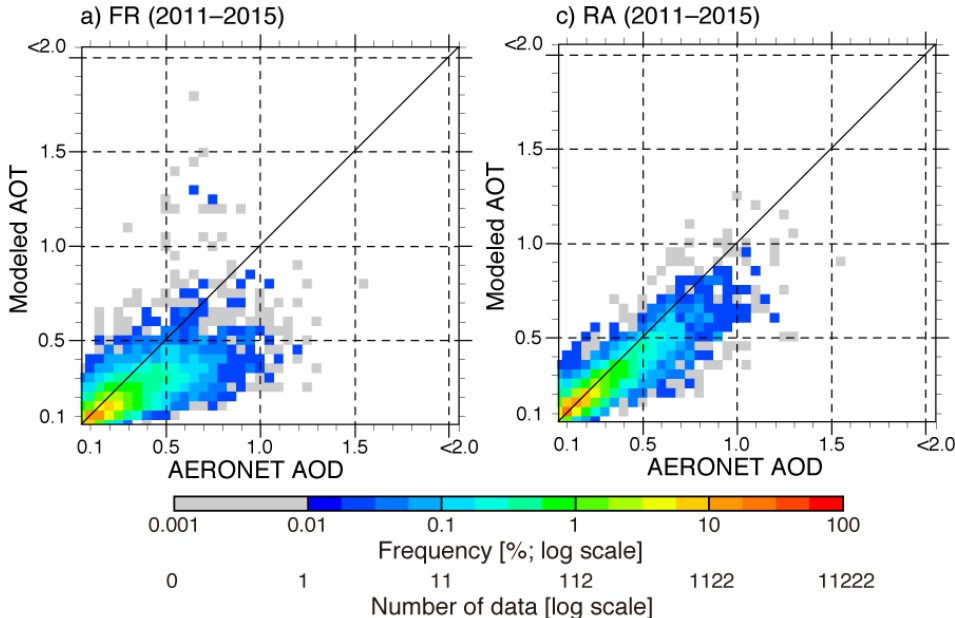

**Figure 14. Scatter plots of (a) FR AODs and (b) RA AODs versus monthly averaged AERONET AODs for the whole reanalysis period (2011–2015).**





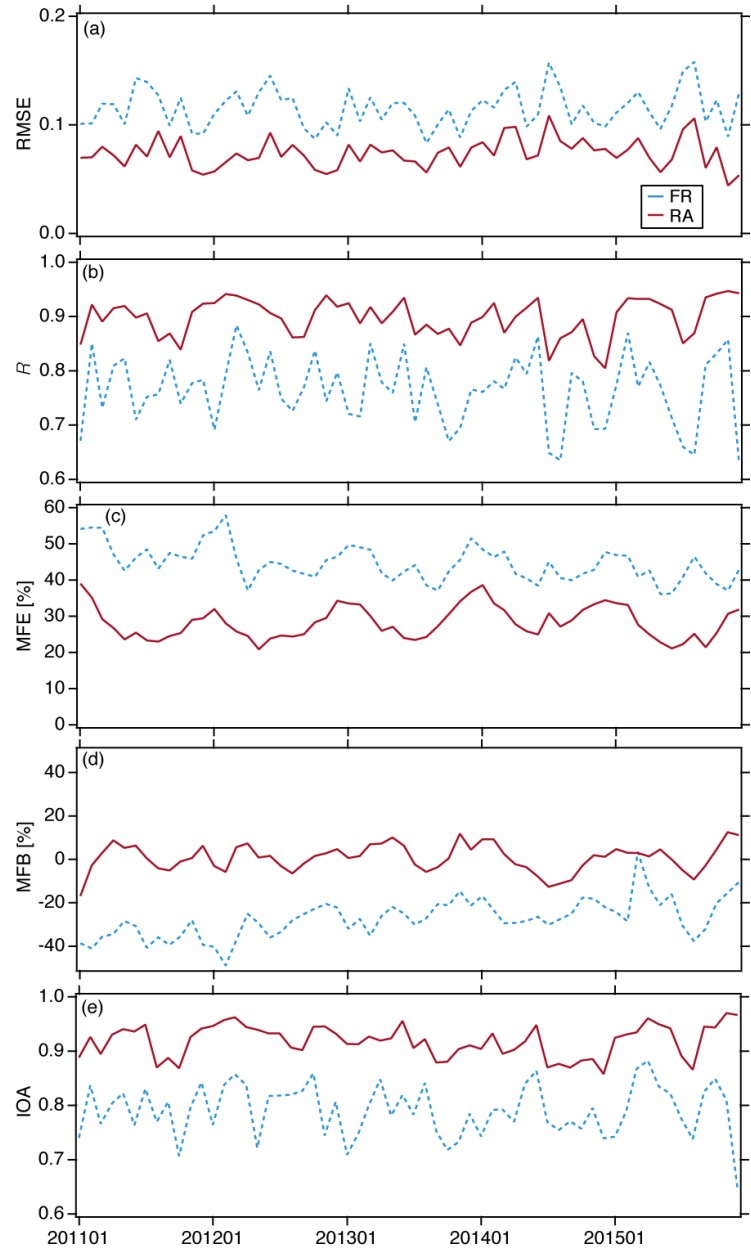

**Figure 15.** Time series of (a) RMSE, (b) *R*, (c) MFE, (d) MFB, and (e) IOA for FR (blue) and RA (red), validated by monthly averaged AERONET AODs. The number of AERONET sites used at each time point for these calculations is shown by the thick black line in Fig. 4.

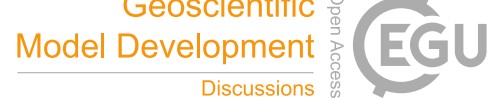



**Figure 16. Maps of statistical measures of monthly AERONET AODs versus RA AODs. (a) RMSE, (b) *R*, (c) MFE, (d) MFB, (e) IOA, and (f) number of data.**





**Figure 17: Same as Fig. 16, except versus FR AODs.**





**Figure 18: Time series of monthly averaged AERONET and modeled AODs. The AERONET, FR, and RA AODs are shown with black, blue, and red lines, respectively. Gray shading denotes standard deviations of the AERONET AODs. Monthly averages of NRL-UND MODIS AODs over each AERONET site are shown by circles. RMSE, *R*, MFE, MFB, and IOA values for RA are also shown. Locations of sites are shown by red circles in Fig. 3. Sites marked by # are included among the sites selected for validation of the ICAP-MME (Sessions et al., 2015).**