# Peer review of "JRAero: the Japanese Reanalysis for Aerosol v1.0"

_Geoscientific Model Development, 2017_

## Referee Comment (RC1) · T. Dai (Referee) · 19 Apr 2017

General Comments: This manuscript describes the development and evaluation of a new global aerosol reanalysis system in Japan. The assimilation system is based on 2D-var method, and the assimilated observation is the corrected MODIS AOD. The reanalysis results are evaluated with both the MODIS AOD and the independent AERONET AODs. The results are interesting and valuable to aerosol data assimilation studies. The manuscript is scientifically sound, original, well written and concise. I recommend accepting it after minor revision as indicated below.

Specific Comments: 1. P5 L3 says the forward model forecast aerosol volume mixing ratio, however the P8 L30 says the AGCM receives mass concentrations from the forward aerosol model. I confuse which is correct. 2. P11 L3 KK should be K. and

how do you calculate the H and HT in the system? 3. P11 L20 how to construct the local regions in the assimilation system? And how to perform the analysis calculation independently in the system? Do you only have one analyzed variable (AOD) in each independent analysis? Please explain more about this. 4. Fig 5. In the FR experiment, do you run the model without restart every six hours? Do you integrate the model for five years one time? If so, does the frequently restart in the RA experiment affect the simulated results? Are the modeled results same with and without frequently restart? 5. P15 L5-10 Firstly, you said the dust particles were increased by assimilation for the Sahel, and you also said the dust particles were decreased for the Mediterranean Sea. Which is right? Could you explain it more? 6. P17 L9 December 2016? You experiments do not include the year 2016. 7. P18 L28 Fig.18g should be Fig. 18a. 8. The formula A2 is wrong.
* * *

---

## Referee Comment (RC2) · Anonymous Referee #2 · 21 Apr 2017

This study developed a global aerosol data assimilation system based on a global aerosol transport model and a 2-dimensional variational data assimilation method. Validations for the reanalysis data were conducted and suggested that the accuracy of the reanalysis data were much higher than the free run of model. I think this paper is valuable for providing an efficient way to obtain high quality reanalysis AOD data through degrading the assimilation system from 3-dimensional to 2-dimensional. I also like the detailed description for the aerosol transport model, the data assimilation system, the observation data, and the set-up of the reanalysis for the global aerosol data assimilation system. I recommend it publish as a technical paper after addressing the following comments. 1. P8, L19-21. The sulfate chemistry in MASINGAR mk-2 includes seven gas-phase reactions and two aqueous-phase reactions. Recently, some studies suggested the CTM model generally underestimated the sulfate, which may be caused by

missing some key heterogeneous chemistry reactions in the model. Could you provide some discussions for this issue related to your sulfate modeling in MASINGAR mk-2. 2. P12, L3. The horizontal error correlation length L is set to 200 km. This may be small for the coarse model resolution of TL159. Could you give more explanation for that? Is that setting related to the localization scale showed in P11, L20-24? 3. P12, L27-28. Please provide more information for the observation error covariance matrix because it is a key issue for the quality of the reanalysis data. Are the standard deviations of the observation errors uniform over the whole model domain? 4. P13, L30-31. Meteorological nudging was performed for the AGCM. What variables in the AGCM were nudged? Is the nudging conducted at the same time step as the AOD data assimilation? 5. P14, L23-25. Sensitivity experiment was conducted through reducing the background error covariance and the chi-square value in the sensitivity experiment was shown. I am interesting in the impact of the change of the background error covariance on the accuracy of the reanalysis AOD data. Could you provide some results for that? 6. P19, L23-24. There is a statement for improving the 6-24h forecast. But I have not seen the experiment for the 24h forecasting.

---

## Referee Comment (RC3) · Anonymous Referee #3 · 26 Apr 2017

This study describes the production of an aerosol reanalysis for the period 2011-2015 through the assimilation of quality assured MODIS observations into the MASINGAR mk-2 global aerosol transport model. The data assimilation scheme used is a 2D-Var method. The paper is well written. Authors describe with clarity the different components of the system: the aerosol transport model, the assimilation method, the observations and the observation operator. The quality of assimilation system is evaluated through internal checks based on analysis and first guess departures from assimilated observations, while the quality of the reanalysis product is evaluated with independent observations, and compared to a control experiment which has been run without data assimilation. I think that the paper is suitable for publication after addressing the following comments.

Specific comments:

1) I am concerned about your background error covariance (BEC) matrix, and it would be good if could add some more comments about them:

a) Can you please justify your choice of using in equation 26 a normalized temporal standard deviation in AOD? Other studies have estimated background error covariances using independent observations or difference of forecasts at different lead times.

b) Can you please justify your choice of expressing the flow-dependent component of the BECs by the forecast AOD (P12, L11-12), and comment on possible drawbacks?

c) Your background error covariances are quite large, in particular (by construction) for high simulated AOD values. Are you not this way over-fitting too much the assimilated observations?

d) Studies by Rubin and colleagues have shown the importance of flow-dependent BECs in aerosol data assimilation, and how ensemble methods can best estimate the temporal evolution of the background error covariances. As you say in the section on "Future directions", you plan to use better BECs in the future, and I think that you should mention this point also when you describe your assimilation method, in the conclusions, and in the abstract.

2) Could you show or comments on whether your analysis is smooth in space and time throughout the day?

a) Given that observations are really assimilated in a given location at most once a day, your reanalysis could have jumps from the one time step where observations are present to the others. If so, you could mention in your conclusions, and in the future outlook, that observations in a future reanalysis should actually have a good coverage. Regarding this issue, Lunch et al. (2016) showed the importance of model tuning in particular when there are areas not covered by assimilated observations, which are therefore highly impacted by the model first guess.

b) When using localization it is important that analyses in neighbour regions share

assimilated observations. How do you deal with this aspect when you divide the model space in local regions?

3) It might be good to stress a bit more that assimilated MODIS observations are not an independent set of observations to validate the reanalysis (e.g., in the abstract P1, L12-15, in the introduction P4, L14-15, . . .), but they can be used to perform valuable sanity checks on the assimilation system.

4) P12, L27: Can you justify the use of a diagonal observation error covariance matrix?

5) P13, L18: Don't you think that it would be more correct to use in the validation AERONET AODs which are the closest in time to model results, without doing any averaging of them?

6) P14, L23-27: What about the accuracy of the analysis when decreasing the BECs? Rather than persistent overestimation of the observation error, could not be that your current BECs simply do not describe well the structures of your background model errors?

7) Could you also add in Figure 6 the analysis increments (analysis minus first guess) and comment on them? This would allow you to identify better local systematic corrections made by the observations and hence discuss model bias, while the differences of Figure 6c (analysis minus free run) are affected also by corrections that might happen somewhere else and are transported.

8) P15, L26 and P19, L24: I think that you can only verify with a certain degree of independence your 6h forecast, and not a forecast up to 24h. Don't you produce a 6h forecast from each analysis step?

9) As you show in P15, L34 and P16, L1, the mean bias and the MFB can have a different sign. Therefore in the discussions in section 3.3.3 should you not add the value of the mean bias, or use the wording "mean fractional bias" when commenting on the MFB values at the various stations?

10) The first few months of simulations (clearly a spin up period for the data assimilation) could have been removed when estimating the statistics reported in the various tables...

Technical corrections:

1) P3, L33: The NASA aerosol reanalysis is called Modern Era Retrospective analysis for Research and Applications Aerosol Reanalysis (MERRAero)

2) P5, L7: Please change the position of the word "respectively" to avoid ambiguity

3) P7, L5: Please change "land-cover factors vegetation" with "land-cover factors for vegetation"

4) P7, L8-L13: Please change "grid(s)" with "grid cell(s)"

5) P7, L21: Please add the air density and the gravitational constant which are also present in equation 9

6) P7: Please use a consistent symbol for particle diameter $d\_s$ or $D\_s$ between equation 9 and 10

7) P11, L3: Please remove one of the K in equation 20

---

## Referee Comment (RC4) · J. Reid (Referee) · 16 May 2017

This is a very comprehensive and well written paper on the JMAero. I don't have too much to add to the other reviews. The task at hand is straightforward: Provide an overview of the model components and provide verification. At least in regard to aerosol optical thickness they have done so admirably. I could follow what they were doing quite well, the figures were well done and they provide comprehensive error stats. One minor thing that needs mentioning, is that the verification really is toward AOT. When NRL composed its reanalysis paper (Lynch et al., 2016), we called it an AOT reanalysis even though it is a full 4 dimensional aerosol simulation because quite frankly AOT was all we could verify against. How these things pan out for surface concentrations is another kettle of fish, and perhaps the authors should note that. Indeed, not only is modeling surface concentrations (or concentrations at any level) difficult,

but finding appropriate verification data is even more difficult. I don't hold the authors accountable to the honest facts, but they should mention it. Similarly, the JMAero does multi-bin particle size distributions (8 bins per specie), but they do not discuss at all how these bins interact, or if at the end of the day it buys them anything as al verification projects onto AOT metrics. This is ok of the paper is about AOT, but that should simply be stated up front in the abstract, introduction and conclusions. Other than this I have a series of equally minor comments that the authors might find helpful. Be well, Jeffrey S. Reid, US Naval Research Laboratory.

Abstract, line 14; Please be clear on an r of 0.96 against the assimilated data. Also, this is just a personal preference, I tend to prefer r2 to r because r2 represents the fraction of variance explained. The authors are of course free to present how they wish, but too often r is used to put a happy face on things. If you want the best of both worlds, you can present bias and rms deviation separately

Please be clear when you refer to "size" if you mean radius or diameter. For example, (Page 6, Line 22) states size ranges from 0.2 to 20 um, but then says the dry radii are from 0.136 to 8.5 (line 23), which implies the original numbers were diameter. Traditionally aerosol science is in cgi units diameter, but it certainly is up to you. Please keep it consistent throughout the paper.

A little more discussion on where secondary OC comes from would help me understand the model better. As a by the way "mk-2 includes production from terpene" is stated on page 8, line 13. In the context of the paragraph in its isolation it is a bit of a non-sequitur. In just a couple of sentences can you please lay out how all primary and secondary POM production is handed with references? Also, no reference is given for the source function of primary POM or BC.

Can you please elaborate a little more on the paragraph starting Page 8, line 28 on how the coupling between AGCM and MASINGAR mk-2? For example what are the timescales of exchange? Are they run at exactly the same resolution? Is data assimilation between meteorology and aerosol particle handed at the same time or are aerosol particles handled after the fact?

On Section 2.2 (Data assimilation). Just a couple of things I am unclear about. First, how does JMAero handle the situation where AOT Obs say there should be a major event, and it is not at all in the model. This happens frequently due to a multitude of mesoscale forcing phenomenon or biomass burning. At NRL we use a climatology, and at GMAO they use the local displacement ensemble.

Section 3.3.2, (Page 13 line 13). The papers describe the AERONET AOPs, but really it is just the AOT that are being used. AOPs implies the inversion products I think.

Page 13, line 29. Listing of vertical levels is ambiguous, Are those the tops of the layers, or the layer thicknesses?

Page 15, line 5: To be fair, the AOT values are not that good either. It is hard to determine who is right when it comes to sea salt...

Page 16, line 24 "70.0% of the deviations exceeded 0" I assume you then mean positive deviations? But then you said that overall the model is negatively biased. You might want to double check the language here.

Page 17, lien 21: Again, don't beat yourself up on Beijing. That and Kanpur have the worst performance in all global models (Sessions et al., 2105). This is a place where 2- and 3 d var is bound to fail. Need EnKF to make it work (http://onlinelibrary.wiley.com/doi/10.1002/2016JD026067/abstract). This I think is different from the overall low bias problem.

Page 18, Line 33: This is because by nature the highest AOT events also have a lot of spatial variability and consequently it gets filtered out in the QA process.

---

## Author Comment (AC1) · 13 Jun 2017

General Comments: This manuscript describes the development and evaluation of a new global aerosol reanalysis system in Japan. The assimilation system is based on 2D-var method, and the assimilated observation is the corrected MODIS AOD. The reanalysis results are evaluated with both the MODIS AOD and the independent AERONET AODs. The results are interesting and valuable to aerosol data assimilation studies. The manuscript is scientifically sound, original, well written and concise. I recommend accepting it after minor revision as indicated below.

We thank a reviewer for careful reading our manuscript and for giving useful comments. We have deliberately and considered your comments. We believe that we have made adequate corrections and answers to your comments. In revised manuscript, the changes are highlighted by yellow markers.

Specific Comments:

1. P5 L3 says the forward model forecast aerosol volume mixing ratio, however the P8 L30 says the AGCM receives mass concentrations from the forward aerosol model. I confuse which is correct.

We unified to "mixing ratio".

2. P11 L3 KK should be K. and how do you calculate the H and HT in the system?

We corrected Eq. (20). As mentioned Line 31 in Page 10, H and HT is the interpolation into observation space and its adjoint (transpose). In the present experiment, since we used the observations re-gridded into the model space, H and HT become the unit matrix.

3. P11 L20 how to construct the local regions in the assimilation system? And how to perform the analysis calculation independently in the system? Do you only have one analyzed variable (AOD) in each independent analysis? Please explain more about this.

In the present system, each element of $\delta\boldsymbol{\tau}^a$ is solved independently (see Eq. (19)). In the other word, the analysis increment of AOD at each grid is calculated independently.

Yes. Each independent analysis solves one analyzed variable at one grid, but uses observed AOD ($\boldsymbol{\tau}^o$), the forecast AOD ($\boldsymbol{\tau}^f$), and background and observation errors (included in **K**) in the local region.

We modified the texts as follows:

"We introduced a localization technique used in LETKF to the system that divides the model space into local regions using a prescribed localization scale. The localization technique solves the analysis increment of AOD at each model grid with observations included in the local region independently (see Eq. (19)), reduces spurious error covariance with distance and enables parallel processing to be used to reduce computational cost."

4. Fig 5. In the FR experiment, do you run the model without restart every six hours? Do you integrate the model for five years one time? If so, does the frequently restart in the RA experiment affect the simulated results? Are the modeled results same with and without frequently restart?

In the FR experiment, we integrate the model for each year. The frequency of restart does not affect the model results.

5. P15 L5-10 Firstly, you said the dust particles were increased by assimilation for the Sahel, and you also said the dust particles were decreased for the Mediterranean Sea. Which is right? Could you explain it more?

The Sahel is the south of the Sahara Desert. The negative difference of dust over the Mediterranean Sea means that the model overestimated the dust over the Sahara Desert (the north of the Sahel), while the dust from the Sahel was underestimated by the model. We modified the text as follows:

"The negative difference around the Mediterranean Sea indicating that the model overestimate dust particles transported from the Sahara Desert."

6. P17 L9 December 2016? You experiments do not include the year 2016.

We corrected.

7. P18 L28 Fig.18g should be Fig. 18a.

We corrected.

8. The formula A2 is wrong.

We corrected.

---

## Author Comment (AC2) · 13 Jun 2017

This study developed a global aerosol data assimilation system based on a global aerosol transport model and a 2-dimensional variational data assimilation method. Validations for the reanalysis data were conducted and suggested that the accuracy of the reanalysis data were much higher than the free run of model. I think this paper is valuable for providing an efficient way to obtain high quality reanalysis AOD data through degrading the assimilation system from 3-dimensional to 2-dimensional. I also like the detailed description for the aerosol transport model, the data assimilation system, the observation data, and the set-up of the reanalysis for the global aerosol data assimilation system. I recommend it publish as a technical paper after addressing the following comments.

We thank a reviewer for careful reading our manuscript and for giving useful comments. We have deliberately and considered your comments. We believe that we have made adequate corrections and answers to your comments. In revised manuscript, the changes are highlighted by yellow markers.

1. P8, L19-21. The sulfate chemistry in MASINGAR mk-2 includes seven gas-phase reactions and two aqueous-phase reactions. Recently, some studies suggested the CTM model generally underestimated the sulfate, which may be caused by missing some key heterogeneous chemistry reactions in the model. Could you provide some discussions for this issue related to your sulfate modeling in MASINGAR mk-2.

I agree with you. The missing of the heterogeneous chemistry reactions is one of the major source of the error (the low bias) in the reanalysis product. It is particularly true in the megacities. For instance, Zhang et al. (2015) evaluated the impacts of heterogeneous chemistry with regional CTM in eastern and central China (urban and industrialized area) and indicated the significant role of heterogeneous chemistry in regional haze (PM2.5) formation. This partly explains the negative biases at megacities in industrializing countries found in the comparison with the AERONET AOD (Section 3.3.2). We added the following texts in the Section 3.3.2 to discuss sources of the negative biases.

"Zhang et al. (2015) evaluated the impact of heterogeneous chemistry with regional chemical transport model in eastern and central China (urban and industrialized area of China), and suggested the significant role of heterogeneous chemistry in regional haze formation. While the current version of MASINGAR mk-2 includes the nine gas-phase and two aqueous-phase reactions of the sulfate

chemistry, the implementation of the heterogeneous chemistry reactions is under development. The missing of the heterogeneous chemical productions may partly explain the negative bias."

We misstated the number of the sulfate chemical reactions included in the current MASINGAR mk-2. We corrected.

"it includes nine gas-phase reactions and two aqueous-phase reactions."

Zheng, B., Zhang, Q., Zhang, Y., He, K. B., Wang, K., Zheng, G. J., Duan, F. K., Ma, Y. L. and Kimoto, T.: Heterogeneous chemistry: a mechanism missing in current models to explain secondary inorganic aerosol formation during the January 2013 haze episode in North China, Atmos. Chem. Phys., 15(4), 2031–2049, doi:10.5194/acp-15-2031-2015, 2015.

2. P12, L3. The horizontal error correlation length L is set to 200 km. This may be small for the coarse model resolution of TL159. Could you give more explanation for that? Is that setting related to the localization scale showed in P11, L20-24?

Yes. Our setting about both the horizontal error correlation length and the localization scale are based on results from Zhang et al. (2008, JGR). They plotted spatial correlation between MODIS observation and their model results as a function of distance (Fig. 7 of Zhang et al. (2008, JGR)), and found that the SOAR function (Eq. (25) in our manuscript) with L = 200 km fit the correlation and the correlation become less than 0.05 when the distance is more than 1000 km. We set the horizontal error correlation length and the localization scale to 200 and 1000 km, respectively, following their results. They used their model with $1° \times 1°$ horizontal resolution to calculate the correlation. Our model resolution of TL159 (about $1.1° \times 1.1°$) is almost the same and the setting of the horizontal error correlation length is reasonable. We unified the explanations about the horizontal error correlation length and add additional texts as follows:

"Zhang et al. (2008) calculated the spatial correlation between satellite observations and model forecasts ($1° \times 1°$ horizontal resolution) as a function of distance and found that the SOAR with $L$ set to 200 km can fit the correlation and when the distance was more than 1000 km, the spatial correlation decreased to less than 0.05. On the basis of their results, we set the localization scale and the horizontal error correlation length to 1000 and 200 km, respectively."

3. P12, L27-28. Please provide more information for the observation error covariance matrix because it is a key issue for the quality of the reanalysis data. Are the standard deviations of the observation errors uniform over the whole model domain?

No. We used quantitative uncertainty estimation for each data point provided by the NRL-UMD MODIS AOD product. The error estimation includes the representativeness error based on variability of the L2 dataset and the observation error estimated empirically. We added the more details of the observation error as follows:

"We assumed that the observation error covariance matrix ($\mathbf{R}$) was diagonal and assigned uncertainty of AOD provided by the NRL-UMD MODIS AOD product to the diagonal component of the observation error covariance matrix. The uncertainty includes empirical estimation of observation error and representativeness error based on variability of the L2 dataset (Zhang and Reid, 2006)."

4. P13, L30-31. Meteorological nudging was performed for the AGCM. What variables in the AGCM were nudged? Is the nudging conducted at the same time step as the AOD data assimilation?

The variables used for the meteorological nudging are horizontal wind components (longitudinal and latitudinal) and air temperature. The nudging forcing is applied at each timestep of the integration by temporary interpolating the variables by linear interpolation. We added these descriptions of the meteorological nudging to the revised manuscript.

"The nudging scheme is applied to the horizontal wind components and air temperature. The nudging term is applied at each time step of the integration by temporary interpolating the variables by linear interpolation."

5. P14, L23-25. Sensitivity experiment was conducted through reducing the background error covariance and the chi-square value in the sensitivity experiment was shown. I am interesting in the

impact of the change of the background error covariance on the accuracy of the reanalysis AOD data. Could you provide some results for that?

Both RA (analyzed) and FG (first guess) from the sensitivity experiment in which the background error covariances were uniformly decreased by 60% shows worse agreement with the MODIS AOD than the main experiment. RMSD (correlation coefficient) for RA and FG during 2011–2012 was degraded from 0.05 (0.93) and 0.098 (0.73) to 0.07 (0.87) and 0.10 (0.71), respectively. The independent validation with the AERONET AOD also shows that the sensitivity experiment obtained worse scores comparing with the main experiment. For chi-square value, the sensitivity experiment shows larger variation compared with the main experiment (0.027 versus 0.0059). These results imply that although the background and observation errors were persistently overestimated, the balance between background and observation errors was well-balanced and stable.

"Both RA and FG from the additional experiment obtained worse scores in the validations with MODIS and AERONET AODs than the standard experiment. For the $\chi^2$ value, the additional experiment shows much larger variation (standard deviation). These results imply that although there were the persistent overestimates of background and observation errors, they were well-balanced and stable in the standard experiment."

6. P19, L23-24. There is a statement for improving the 6-24h forecast. But I have not seen the experiment for the 24h forecasting.

We employed the MODIS observation as the assimilation data. MODIS is onboard Low Earth Orbit (LEO) satellites and make observation on the same region once a day. What we want to mean by "short-term (6–24 hour) forecasting" is a short-term model simulation after the analysis to the next analysis (when the next MODIS observation is available) in a certain region. We corrected the text to "the short-term forecasting until the next analysis (until the next MODIS AOD is available)".

---

## Author Comment (AC3) · 13 Jun 2017

This study describes the production of an aerosol reanalysis for the period 2011-2015 through the assimilation of quality assured MODIS observations into the MASINGAR mk-2 global aerosol transport model. The data assimilation scheme used is a 2D-Var method. The paper is well written. Authors describe with clarity the different components of the system: the aerosol transport model, the assimilation method, the observations and the observation operator. The quality of assimilation system is evaluated through internal checks based on analysis and first guess departures from assimilated observations, while the quality of the reanalysis product is evaluated with independent observations, and compared to a control experiment which has been run without data assimilation. I think that the paper is suitable for publication after addressing the following comments.

We thank a reviewer for careful reading our manuscript and for giving useful comments. We have deliberately and considered your comments. We believe that we have made adequate corrections and answers to your comments. In revised manuscript, the changes are highlighted by yellow markers.

Specific comments:

1) I am concerned about your background error covariance (BEC) matrix, and it would be good if could add some more comments about them:

a) Can you please justify your choice of using in equation 26 a normalized temporal standard deviation in AOD? Other studies have estimated background error covariances using independent observations or difference of forecasts at different lead times.

We got an idea from the NMC method for the estimate of BECs in the current version. In the NMC, variations of forecasts at different lead times are used. In this study, we used variations of forecasts at different times ($\hat{\sigma}_{FR}$), because, for the reanalysis use, we do not perform the forecasts with different lead times. However, comparing with variations of forecasts with different lead times, distribution of variation forecasts at different times becomes to be smoothen and makes shape of aerosol plumes obscured. This is the reason that we used the estimate expressed by Eq. (26). We added the following text and use "the horizontal structure of AOD field at the assimilation time" instead of "the flow-dependent structure of AOD" to convey our intentions more correctly.

"The fraction on the right-hand side of Eq. (26) indicates that the standard deviation is normalized by the mean value. The standard deviation derived from 31-day AODs leads to relatively smooth variation fields. To introduce the horizontal structure of AOD field at the assimilation time into the background error covariance, the normalized standard deviation is multiplied by the forecast AOD."

b) Can you please justify your choice of expressing the flow-dependent component of the BECs by the forecast AOD (P12, L11-12), and comment on possible drawbacks?

Thank you for your question. Please refer the previous reply (reply to the comment #1a) for question about "flow-dependent", and I would discuss about the drawbacks in the reply to the next comment (reply to the comment #1c).

c) Your background error covariances are quite large, in particular (by construction) for high simulated AOD values. Are you not this way over-fitting too much the assimilated observations?

Large BEC for high simulated AOD values might be a common problem in the aerosol assimilation even with the NMC and ensemble based-methods, because high simulated AOD often leads larger variation (spread).

The chi-square test and the sensitivity experiment in which the background error covariances were uniformly decreased by 60% shows that although the background and observation errors were persistently overestimated, the balance between background and observation errors was well-balanced and stable (see reply to the comment #6). The sanity test with the MODIS AOD and examination for the first guess (FG) confirm that the analyses caused few over-fitting. We added the texts in the revised manuscript as follows:

"Both RA and FG from the additional experiment obtained worse scores in the validations with MODIS and AERONET AODs than the standard experiment. For the $\chi^2$ value, the additional experiment shows much larger variation (standard deviation). These results imply that although there were the persistent overestimates of background and observation errors, they were well-balanced and stable in the standard experiment."

One of possible drawbacks for the estimate of BECs is the analyses at megacities. Please refer reply to the next comment (comment #1d) for this point.

In the current version of JRAero, the background error was in proportion to the forecast AOD (i.e., first guess AOD; see Eq. (26)). This means that the background errors where the model did not predict aerosols became so small. Therefore, in that situation, the assimilation could not reproduce the major aerosol event because of the small background error (also see reply to the comment #5 of reviewer #4). This should be another possible drawback. We added discussion about this situation in Section 5 (Future direction) as follows:

"In the current version, the background error was in proportion to the forecast AOD (Eq. (26)), and became small where the model did not predict aerosols. Therefore, the analysis could not reproduce aerosol events that satellites observed but the model failed to predict (e.g., dust storms and biomass burning) due to the small background error. The ensemble-based estimate of the background error considering uncertainty in emissions will bring better analysis for this situation."

d) Studies by Rubin and colleagues have shown the importance of flow-dependent BECs in aerosol data assimilation, and how ensemble methods can best estimate the temporal evolution of the background error covariances. As you say in the section on "Future directions", you plan to use better BECs in the future, and I think that you should mention this point also when you describe your assimilation method, in the conclusions, and in the abstract.

Thank you for your suggestion. We added the following text and use "the horizontal structure of AOD field at the assimilation time" instead of "the flow-dependent structure of AOD" to convey our intentions more correctly, as answered in the previous comment.

Better estimate of BECs is our important direction to next version of the analysis and ensemble-based methods are quite powerful to obtain the flow-dependent BECs. We added the following texts that mention that the ensemble-based estimate of background error covariance has the possibility to overcome problems included in the current version of the reanalysis referring the most recent results by Rubin et al. (2017) in Section 3.3.2 and Section 5 (also see reply to comment #10 of reviewer #4).

Section 3.3.2: "The probability of a successful retrieval can be reduced during high-AOD events (Lynch et al., 2016); thus, fewer available satellite observations over megacities during high-AOD events may also account for the negative biases in RA AODs. Rubin et al. (2017) applied an ensemble-based assimilation method to NAAPS and found that flow-dependent error covariances estimated by ensemble simulations utilized the AERONET AOD efficiently and brought better analyses at Beijing and Kanpur. Sophistication of the background error covariance and assimilation of additional observations have the potential to improve the analyses at the megacities."

Future directions: "At megacity and mountain sites, assimilation provided limited improvement, and positive and negative biases remained in the reanalysis. A plausible reason is the coarse model resolution, which is insufficient to resolve high-AOD events around megacities and local terrain effects in mountainous areas. Therefore, we plan to rerun the reanalysis with a finer resolution and check the performance of the model. Re-examination of the background error by an ensemble-based method (Yumimoto et al., 2016) and assimilation of additional observations (e.g., the AERONET AOD) have the potential to improve the analyses at the megacity sites (Rubin et al., 2017)."

Rubin, J. I., Reid, J. S., Hansen, J. A., Anderson, J. L., Holben, B. N., Lynch, P., Westphal, D. L. and Zhang, J.: Assimilation of AERONET and MODIS AOT observations using Variational and Ensemble Data Assimilation Methods and Its Impact on Aerosol Forecasting Skill, J. Geophys. Res. Atmos., doi:10.1002/2016JD026067, 2017.

This paper constitutes a comprehensive report on the current version of JRAero. From the validation studies, we found that the estimate of BECs seems to be large and should be improved. However, that is beyond the scope of this paper. Therefore, we added more detailed discussion about the update of the BECs in the Evaluation results and Future directions rather than the Conclusions and the Abstract.

2) Could you show or comments on whether your analysis is smooth in space and time throughout the day?

a) Given that observations are really assimilated in a given location at most once a day, your reanalysis could have jumps from the one time step where observations are present to the others. If

so, you could mention in your conclusions, and in the future outlook, that observations in a future reanalysis should actually have a good coverage. Regarding this issue, Lunch et al. (2016) showed the importance of model tuning in particular when there are areas not covered by assimilated observations, which are therefore highly impacted by the model first guess.

Your suggestion is important with aspect of temporal continuity of reanalysis product (the non-diagonal BECs prevent the reanalysis from spatial jumps caused by the spatial limited MODIS AOD due to clouds and snows). We added the following texts in the revised manuscript.

Future direction: "The limited temporal coverage of MODIS AOD (once a day) might cause temporal jumps or discontinuities in the reanalysis. Lunch et al. (2016) suggested that the importance of model tuning in particular when there are areas not covered by assimilated observations. To obtain a better coverage of assimilation data is important for the future development. Smoother techniques (e.g., 4-dimensional variational method (Yumimoto and Takemura, 2013; 2015) and ensemble Kalman smoother (Schutgens et al. (2012)) are also useful."

b) When using localization it is important that analyses in neighbour regions share assimilated observations. How do you deal with this aspect when you divide the model space in local regions?

The local region is set to each model grid. The analysis increment of AOD ($\delta\boldsymbol{\tau}^a$) at each model grid is solved independently with observations (more correctly, innovations ($\boldsymbol{\tau}^o - \mathbf{H}_I\boldsymbol{\tau}^f$)) included in the local region (see Eq. (19)). Also see reply to comment #3 by the reviewer #1.

We modified the texts as follows:

"We introduced a localization technique used in LETKF to the system that divides the model space into local regions using a prescribed localization scale. The localization technique solves the analysis increment of AOD at each model grid with observations included in the local region independently (see Eq. (19)), reduces spurious error covariance with distance and enables parallel processing to be used to reduce computational cost."

3) It might be good to stress a bit more that assimilated MODIS observations are not an independent set of observations to validate the reanalysis (e.g., in the abstract P1, L12-15, in the introduction P4, L14-15, . . .), but they can be used to perform valuable sanity checks on the assimilation system.

Thank you for your suggestion.

We added the texts in the revised manuscript.

Abstract: "Comparisons with MODIS AODs showed that the reanalysis showed much better agreement than the free run (without assimilation) of the aerosol model and improved under- and overestimation in the free run, thus confirming the sanity of the data assimilation system."

Introduction: "Section 3 focuses on the sanity check and evaluation of the reanalysis product with MODIS AOD and independent observation data."

Tittle of the Section 3.2: "Sanity check by MODIS AODs"

4) P12, L27: Can you justify the use of a diagonal observation error covariance matrix?

Ideally, we should utilize correlation in the observation error covariance. However, accurate estimate of the correlation remains challenging (even in NWP assimilation). We need to put a lot of effort into this problem both for production of reanalysis and aerosol forecasting.

We add some texts in the revised manuscript.

"We assumed that the observation error covariance matrix ($R$) was diagonal because of the difficulty of accurate estimate of the correlation and assigned uncertainty of AOD provided by the NRL-UMD MODIS AOD product to the diagonal component of the observation error covariance matrix. The uncertainty includes empirical estimation of observation error and representativeness error based on variability of the L2 dataset (Zhang and Reid, 2006)."

5) P13, L18: Don't you think that it would be more correct to use in the validation AERONET AODs which are the closest in time to model results, without doing any averaging of them?

The best way is to compare observed AOD with modeled AOD at the observed time. However, that is unfeasible, because the temporal resolution of the AERONET AOD is much finer than that of model output. Additionally, the AERONET AODs include fine temporal variations (less than 1 hour) that the global model cannot reproduce. The aim of the reanalysis is to provide 6-hourly, daily and monthly aerosol fields, not reproduce the fine temporal variations. Therefore, we averaged the AERONET AODs into 1-hour bin, and compared with the model results.

6) P14, L23-27: What about the accuracy of the analysis when decreasing the BECs? Rather than persistent overestimation of the observation error, could not be that your current BECs simply do not describe well the structures of your background model errors?

Both RA (analyzed) and FG (first guess) from the sensitivity experiment in which the background error covariances were uniformly decreased by 60% shows worse agreement with the MODIS AOD than the main experiment. RMSD (correlation coefficient) for RA and FG during 2011–2012 was degraded from 0.05 (0.93) and 0.098 (0.73) to 0.07 (0.87) and 0.10 (0.71), respectively. The independent validation with the AERONET AOD also shows that the sensitivity experiment obtained worse scores comparing with the main experiment. For chi-square value, the sensitivity experiment shows larger variation compared with the main experiment (0.027 versus 0.0059). These results imply that although the background and observation errors were persistently overestimated, the balance between background and observation errors was well-balanced and stable.

"Both RA and FG from the sensitivity experiment obtained worse scores in the validations by MODIS and AERONET AODs than the main experiment. For the $\chi^2$ value, the sensitivity experiment shows much larger variation (standard deviation) compared with the main experiment. These results imply that although there are the persistent overestimates of background and observation errors, they were well-balanced and stable in the main experiment."

7) Could you also add in Figure 6 the analysis increments (analysis minus first guess) and comment on them? This would allow you to identify better local systematic corrections made by the observations and hence discuss model bias, while the differences of Figure 6c (analysis minus free run) are affected also by corrections that might happen somewhere else and are transported.

We added increments (analysis minus first guess) as Figure 6d, and re-labeled analysis minus free run as difference (Fig. 6c). Comments on them were added in the manuscript.

"Figure 6d exhibits the distribution of the increment (RA AOD minus FG AOD) that is derived from the 5-year average of the modifications by the assimilations (i.e., $\delta\tau^a$ in Eq. (19)). The increment shows lower amplitudes and different distributions in several regions (particularly in the downwind regions of aerosol sources) compared with the difference (Fig. 6c). It is because that the difference is affected by transport of the modifications after the assimilations and the increment (FG AOD) takes into account accumulation of previous assimilations. The effect by the accumulation also appears as much better statistics of FG AOD (e.g., lower root mean square error (RMSD) and mean fractional bias (MFB)) comparing with FR AOD (Fig. 7)."

8) P15, L26 and P19, L24: I think that you can only verify with a certain degree of independence your 6h forecast, and not a forecast up to 24h. Don't you produce a 6h forecast from each analysis step?

We employed the MODIS observation as the assimilation data. MODIS is onboard Low Earth Orbit (LEO) satellites and make observation on the same region once a day. What we want to mean by "short-term (6–24 hour) forecasting" is a short-term model simulation after the analysis to the next analysis (when the next MODIS observation is available) in a certain region. We corrected the text to "the short-term forecasting until the next analysis (until the next MODIS AOD is available)".

9) As you show in P15, L34 and P16, L1, the mean bias and the MFB can have a different sign. Therefore in the discussions in section 3.3.3 (You mean Section 3.3.2?) should you not add the value of the mean bias, or use the wording "mean fractional bias" when commenting on the MFB values at the various stations?

We used "a positive MFB value" and "a negative MFB value" in the discussions in Section 3.3.2 when commenting on the MFB values in the revised manuscript.

10) The first few months of simulations (clearly a spin up period for the data assimilation) could have been removed when estimating the statistics reported in the various tables...

As mentioned in the Subsection 2.4, we made the 3-month spin-up (October–December 2010) of free run and then performed the 5-year reanalysis. The spin-up period was excluded from the validations. However, to make a more perfect validation, we also should make few months of spin-up with data assimilation. In the next version of the reanalysis, we will introduce a spin-up with data assimilation (e.g., few months of spin-up by free run, few months of spin-up with data assimilation, and then reanalysis period).

Technical corrections:

We thank a reviewer for careful reading our manuscript and for giving useful correction. We corrected in the revised manuscript.

1) P3, L33: The NASA aerosol reanalysis is called Modern Era Retrospective analysis for Research and Applications Aerosol Reanalysis (MERRAero)

Now, aerosol variables are included in additional file collections in MERRA-2. (https://disc.gsfc.nasa.gov/datareleases/merra_2_data_release)

2) P5, L7: Please change the position of the word "respectively" to avoid ambiguity

3) P7, L5: Please change "land-cover factors vegetation" with "land-cover factors for vegetation"

4) P7, L8-L13: Please change "grid(s)" with "grid cell(s)"

5) P7, L21: Please add the air density and the gravitational constant which are also present in equation 9

6) P7: Please use a consistent symbol for particle diameter d_s or D_s between equation 9 and 10

7) P11, L3: Please remove one of the K in equation 20

---

## Author Comment (AC4) · 13 Jun 2017

This is a very comprehensive and well written paper on the JMAero. I don't have too much to add to the other reviews. The task at hand is straightforward: Provide an overview of the model components and provide verification. At least in regard to aerosol optical thickness they have done so admirably. I could follow what they were doing quite well, the figures were well done and they provide comprehensive error stats. One minor thing that needs mentioning, is that the verification really is toward AOT. When NRL composed its reanalysis paper (Lynch et al., 2016), we called it an AOT reanalysis even though it is a full 4 dimensional aerosol simulation because quite frankly AOT was all we could verify against. How these things pan out for surface concentrations is another kettle of fish, and perhaps the authors should note that. Indeed, not only is modeling surface concentrations (or concentrations at any level) difficult, but finding appropriate verification data is even more difficult. I don't hold the authors accountable to the honest facts, but they should mention it. Similarly, the JMAero does multi-bin particle size distributions (8 bins per specie), but they do not discuss at all how these bins interact, or if at the end of the day it buys them anything as al verification projects onto AOT metrics. This is ok of the paper is about AOT, but that should simply be stated up front in the abstract, introduction and conclusions. Other than this I have a series of equally minor comments that the authors might find helpful. Be well, Jeffrey S. Reid, US Naval Research Laboratory.

We thank a reviewer for careful reading our manuscript and for giving useful comments. We have deliberately and considered your comments. We believe that we have made adequate corrections and answers to your comments. In revised manuscript, the changes are highlighted by yellow markers.

As the reviewer pointed out, we validated the quality of the reanalysis with only AOD measurements. We added the text in the Abstract, Introduction, Conclusions, and Future directions as follows:

Abstract: "This paper describes the aerosol transport model, the data assimilation system, the observation data, and the set-up of the reanalysis and examines its quality with AOD observations."

Introduction: ", and validated its quality with AOD observations."

Conclusions: "A global aerosol data assimilation system was developed based on a global aerosol transport model, MASINGAR mk-2, and a 2-dimensional variational data assimilation method, and validated by AOD measurements."

Future directions: "In the present paper, the reanalysis was validated with only the AOD observations. Further validation is also needed. Comparison with vertical AOD distributions…"

1) Abstract, line 14; Please be clear on an r of 0.96 against the assimilated data. Also, this is just a personal preference, I tend to prefer r2 to r because r2 represents the fraction of variance explained. The authors are of course free to present how they wish, but too often r is used to put a happy face on things. If you want the best of both worlds, you can present bias and rms deviation separately

I stressed the results from comparison with observation that used for the assimilation as follows:

"Comparisons with MODIS AODs that used for the assimilation showed that the reanalysis showed much better agreement than the free run (without assimilation) of the aerosol model and improved under- and overestimation in the free run, thus confirming the sanity of the data assimilation system."

Thank you for your suggestion. I understand your concern. However, in the present paper, we used r throughout the manuscript (main body, appendix, figures, and tables), and need a complete revision to unify into r2. In the next study, we carefully select the statistics to meet the aim of validation.

2) Please be clear when you refer to "size" if you mean radius or diameter. For example, (Page 6, Line 22) states size ranges from 0.2 to 20 um, but then says the dry radii are from 0.136 to 8.5 (line 3), which implies the original numbers were diameter. Traditionally aerosol science is in cgi units diameter, but it certainly is up to you. Please keep it consistent throughout the paper.

Thank you for your useful comment. We use "diameter" in the corrected manuscript.

"For mineral dust particles, the model uses a size bin method that logarithmically divides the particle size range from 0.2 to 20 μm into 10 size classes. The volumetric mean diameters of dry particles in each size bin are 0.271, 0.430, 0.681, 1.08, 1.71, 2.71, 4.30, 6.81, 10.8, and 17.1 μm."

3) A little more discussion on where secondary OC comes from would help me understand the model better. As a by the way "mk-2 includes production from terpene" is stated on page 8, line 13. In the context of the paragraph in its isolation it is a bit of a non-sequitur. In just a couple of sentences can you please lay out how all primary and secondary POM production is handed with references? Also, no reference is given for the source function of primary POM or BC.

MASINGAR mk-2 does not treat secondary organic aerosol production explicitly but represents it implicitly by giving OC amounts produced from terpene. OC produced from terpene is assumed to be secondary organic aerosol and treated as hydrophilic. The emission of terpene is included in the emission inventory.

We added the following text in the revised manuscript.

"Although MASINGAR mk-2 does not calculate secondary organic aerosol production explicitly, this process is represented implicitly by giving OC production from terpene using emission data. Emission amount of terpene is provided by the emission inventory. OC produced from terpene is assumed to be secondary organic aerosol and treated as hydrophilic."

4) Can you please elaborate a little more on the paragraph starting Page 8, line 28 on how the coupling between AGCM and MASINGAR mk-2? For example what are the timescales of exchange? Are they run at exactly the same resolution? Is data assimilation between meteorology and aerosol particle handed at the same time or are aerosol particles handled after the fact?

Thank you for pointing out about the coupling. As the settings for the coupling depend on experiments, we added the detailed explanation about setting for coupling in the Section 2.4 (Experiment setup) rather than Section 2.1 (Overview of MASINGAR mk-2).

In present experiment, spatial resolution of the AGCM is the same to that of MASINGAR mk-2. Therefore, there is no spatial interpolation of the exchange variables. The timescale of exchange is the same to the time step of model (i.e., 900 sec); The AGCM and MASINGAR mk-2 exchange the variables at every time steps. At the aerosol assimilation time, the exchange is performed after the assimilation. This means that the AGCM receives the analyzed aerosol field from MASINGAR mk-2. We revised the Section 2.4 as follows:

"The AGCM has the same spatial resolution and time step to MASINGAR mk-2. The exchange of meteorological and aerosol variables between the AGCM and MASINGAR mk-2 through the coupler is performed every model time steps (i.e., every 900 seconds). At the assimilation step, the exchange follows the assimilation. This means that the AGCM receives the analyzed aerosol fields from MASINGAR mk-2."

5) On Section 2.2 (Data assimilation). Just a couple of things I am unclear about. First, how does JMAero handle the situation where AOT Obs say there should be a major event, and it is not at all in the model. This happens frequently due to a multitude of mesoscale forcing phenomenon or biomass burning. At NRL we use a climatology, and at GMAO they use the local displacement ensemble.

In the current version of JRAero, the background error was in proportion to the forecast AOD (i.e., first guess AOD; see Eq. (26)). This means that the background errors where the model did not predict aerosols became so small. Therefore, in that situation, the assimilation could not reproduce the major aerosol event because of the small background error.

Missing of aerosol sources (particularly for dust storms and biomass burnings) frequently causes the situation. Data assimilation both for initial condition and aerosol sources (i.e., emission inverse modeling; Yumimoto et al., 2008) will result in the better solution to the situation. Other effort should be application of an ensemble-based method with perturbed aerosol emissions.

We added discussion about this situation in Section 5 (Future direction) as follows:

"In the current version, the background error was in proportion to the forecast AOD (Eq. (26)), and became small where the model did not predict aerosols. Therefore, the analysis could not reproduce aerosol events that satellites observed but the model failed to predict (e.g., dust storms and biomass burning) due to the small background error. The ensemble-based estimate of the background error considering uncertainty in emissions will bring better analysis for this situation."

Yumimoto, K., Uno, I., Sugimoto, N., Shimizu, a., Liu, Z. and Winker, D. M.: Adjoint inversion modeling of Asian dust emission using lidar observations, Atmos. Chem. Phys., 8(11), 2869–2884, doi:10.5194/acp-8-2869-2008, 2008.

6) Section 3.3.2, (Page 13 line 13). The papers describe the AERONET AOPs, but really it is just the AOT that are being used. AOPs implies the inversion products I think.

We corrected the manuscript.

7) Page 13, line 29. Listing of vertical levels is ambiguous, Are those the tops of the layers, or the layer thicknesses?

We corrected to the layer thickness (or box height) in the revised manuscript.

8) Page 15, line 5: To be fair, the AOT values are not that good either. It is hard to determine who is right when it comes to sea salt. . .

I agree with you. We need to check carefully where the errors come from. As you concerned, for instance, dust and anthropogenic aerosols could affect total AOD over Atlantic Ocean and northern Pacific Ocean. We corrected the text as follows:

"The distribution of the 5-year averaged increment (RA AOD minus FR AOD) (Fig. 6c) shows that, in general, assimilation increased AODs over the central Pacific Ocean, implying that in FR, MASINGAR mk-2 underestimated sea-salt aerosols."

9) Page 16, line 24 "70.0% of the deviations exceeded 0" I assume you then mean positive deviations? But then you said that overall the model is negatively biased. You might want to double check the language here.

In the figures, we defined the deviation as observed AOD minus modeled AOD. So, the positive deviation means that the model underestimates the observation.

10) Page 17, lien 21: Again, don't beat yourself up on Beijing. That and Kanpur have the worst performance in all global models (Sessions et al., 2105). This is a place where 2- and 3 d var is bound to fail. Need EnKF to make it work (http://onlinelibrary.wiley.com/doi/10.1002/2016JD026067/abstract). This I think is different from the overall low bias problem.

Thank you for your encouragement and suggestion.

As shown in Fig. 9, the model has the constant negative biases in the megacities mainly due to the insufficient anthropogenic emissions, the coarse model resolution and the missing of heterogeneous productions. We added more discussion about the negative biases caused by the model errors (also see reply to comment #1 of the reviewer #2).

"Multi-model inter-comparison studies (Kinne et al., 2006; Sessions et al., 2015) have pointed out that aerosol models having negative biases for high-AOD events is a common problem. Insufficient anthropogenic emissions data and model resolution for megacities are plausible reasons for the negative biases. Zhang et al. (2015) evaluated the impact of heterogeneous chemistry with regional chemical transport model in eastern and central China (urban and industrialized area of China), and suggested the significant role of heterogeneous chemistry in regional haze formation. While the current version of MASINGAR mk-2 includes the nine gas-phase and two aqueous-phase reactions of the sulfate chemistry, the implementation of the heterogeneous chemistry reactions is under development. The missing of the heterogeneous chemical productions may partly explain the negative bias."

The limited improvement by the assimilation is other problem. We have to take some steps about the low biases in the megacities both on the model and the assimilation. In the aspect of assimilation, as the reviewer suggested, the improvement of background error covariance is useful. We added the following texts that mention that the ensemble-based estimate of background error covariance has the possibility to overcome the problem referring the most recent results by Rubin et al. (2017).

Section 3.3.2: "The probability of a successful retrieval can be reduced during high-AOD events (Lynch et al., 2016); thus, fewer available satellite observations over megacities during high-AOD

events may also account for the negative biases in RA AODs. Rubin et al. (2017) applied an ensemble-based assimilation method to NAAPS and found that flow-dependent error covariances estimated by ensemble simulations utilized the AERONET AOD efficiently and brought better analyses at Beijing and Kanpur. Sophistication of the background error covariance and assimilation of additional observations have the potential to improve the analyses at the megacities."

Future directions: "At megacity and mountain sites, assimilation provided limited improvement, and positive and negative biases remained in the reanalysis. A plausible reason is the coarse model resolution, which is insufficient to resolve high-AOD events around megacities and local terrain effects in mountainous areas. Therefore, we plan to rerun the reanalysis with a finer resolution and check the performance of the model. Re-examination of the background error by an ensemble-based method (Yumimoto et al., 2016) and assimilation of additional observations (e.g., the AERONET AOD) have the potential to improve the analyses at the megacity sites (Rubin et al., 2017)."

Rubin, J. I., Reid, J. S., Hansen, J. A., Anderson, J. L., Holben, B. N., Lynch, P., Westphal, D. L. and Zhang, J.: Assimilation of AERONET and MODIS AOT observations using Variational and Ensemble Data Assimilation Methods and Its Impact on Aerosol Forecasting Skill, J. Geophys. Res. Atmos., doi:10.1002/2016JD026067, 2017.

11) Page 18, Line 33: This is because by nature the highest AOT events also have a lot of spatial variability and consequently it gets filtered out in the QA process.

I agree with your point. I added the point in the revised manuscript.
"The NRL-UND MODIS peak AOD values were also lower than the AERONET peak AODs, because larger spatial variability included in the dense aerosol events filtered the high AOD values out through the QA process."